# Aging increases vulnerability to stress-induced depression via upregulation of NADPH oxidase in mice

Jung-Eun Lee[1], Hye-Jin Kwon[1], Juli Choi[1], Ji-Seon Seo[1] & Pyung-Lim Han[1,2,3 ✉]

Brain aging proceeds with cellular and molecular changes in the limbic system. Aging-dependent changes might affect emotion and stress coping, yet the underlying mechanisms remain unclear. Here, we show aged (18-month-old) mice exhibit upregulation of NADPH oxidase and oxidative stress in the hippocampus, which mirrors the changes in young (2-month-old) mice subjected to chronic stress. Aged mice that lack p47phox, a key subunit of NADPH oxidase, do not show increased oxidative stress. Aged mice exhibit depression-like behavior following weak stress that does not produce depressive behavior in young mice. Aged mice have reduced expression of the epigenetic factor SUV39H1 and its upstream regulator p-AMPK, and increased expression of Ppp2ca in the hippocampus—changes that occur in young mice exposed to chronic stress. SUV39H1 mediates stress- and aging-induced sustained upregulation of p47phox and oxidative stress. These results suggest that aging increases susceptibility to stress by upregulating NADPH oxidase in the hippocampus.

[1] Department of Brain and Cognitive Sciences, Ewha Womans University, Seoul 03760, Republic of Korea. [2] Department of Chemistry and Nano Science, Ewha Womans University, Seoul 03760, Republic of Korea. [3] Brain Disease Research Institute, Ewha Womans University, Seoul 03760, Republic of Korea. ✉email: plhan@ewha.ac.kr

Brain aging proceeds through various cellular and molecular changes in the limbic system. Aged brains show increased oxidative stress[1,2], altered expression of genes that regulate oxidative stress[3], and reduced expression of neurotrophins[4,5] in the hippocampus. These aging-dependent changes are closely related to stress hormones[6,7], and overlap with those of stress-induced changes in the brain of young animals[8,9]. Aged people exhibit increased basal serum glucocorticoid (GC) levels[10,11]. Depression patients also exhibit increased basal serum GC levels[12–14]. These reports suggest that the stress-regulating neuroendocrine system is dysfunctional in aged people and patients with depression. These results raise the questions of whether aging advances the course of stress-induced changes, and whether aging-induced changes promote stress-induced depression.

GCs (cortisol in human and corticosterone in rodents) are the key stress hormones released into the circulatory system as a result of hypothalamus-pituitary-adrenal (HPA) axis activation[15,16]. GCs exert their effect through glucocorticoid receptors (GRs), which are expressed at high levels in the limbic system, including the hippocampus[17,18]. GCs upregulate nicotinamide adenine dinucleotide phosphate (NADPH) oxidase and oxidative stress[19,20], and reduce the expression of neurotrophins in vitro[21,22]. Stress-induced upregulation of NADPH oxidase in the hippocampus of mice promotes depression-like behaviors[20]. Furthermore, mice subjected to chronic stress exhibit increased basal serum GC levels and HPA-axis dysfunction[23,24]. Aging-dependent oxidative stress and HPA-axis dysfunction are similar to those induced by chronic stress in human[7], although the detailed mechanism through which aging-dependent changes increase emotional sensitivity to stress is not clearly understood. While recent studies on stress-induced depression have advanced our understanding of stress-induced depression, most such studies have been conducted on young animals. Whether similar mechanisms are involved in the regulation of stress coping and stress-induced depression later in life is unclear.

AMP-activated protein kinase (AMPK) senses and regulates cellular energy homeostasis and redox states[25,26]. AMPK is phosphorylated by upstream kinases CaMK2, TAK1 (transforming growth factor-β-activated kinase 1), and LKB1, whereas phospho-AMPK is deactivated by protein phosphatases PP2A, P2Cα, and Ppm1E[27]. Phospho-AMPK reduces NADPH oxidase expression in endothelial cells and aorta cells isolated from AMPKα2-deficient mice[28], although it is unknown whether a similar AMPK-dependent NADPH oxidase expression occurs in neuronal cells. AMPK activity is reduced in the brain of stress models of depression[29,30]. In contrast, AMPK activation increases phospho-CREB expression[31,32], and improves impaired depressive behaviors[33]. These results suggest that AMPK and factors centered on AMPK have a role in stress-induced pathophysiology in aged brains.

Epigenetic modifications of histone proteins play a role in development of psychiatric disorders including depression[34,35]. Among the epigenetic factors that regulate stress-induced depressive behaviors[36,37], SUV39H1, which is a histone methyltransferase that adds a tri- or dimethyl group to histone-3 lysine 9 (H3K9), is downregulated in the hippocampus after treatment with chronic stress and its reduced expression causes depressive behaviors[38]. Given the pivotal role of epigenetic mechanisms in aging[39], whether SUV39H1 or other epigenetic factors play a role in aging brains is therefore a crucial question.

In the present study, we investigated how aging proceeds with changes of stress-regulating factors and whether aging-dependent changes increase emotional sensitivity to stress.

## Results

**NADPH oxidase mediates aging-induced accumulation of ROS.** Aged mice (18 months of age) had higher basal serum corticosterone levels compared with young (2-month-old) mice. However, aged mice released less corticosterone after exposed to a 2-h restraint relative to young mice (Fig. 1a). Adrenal glands of aged mice were heavier than those of young mice, and corticosterone levels were significantly correlated to increased adrenal gland weight in aged mice (Fig. 1b, c). The hypophysiotrophic factors corticotrophin-releasing hormone (CRH) and vasopressin (AVP) in the paraventricular nucleus of the hypothalamus (PVN) in aged mice were upregulated compared with young mice (Fig. 1d, e). These results suggest that the HPA axis is dysregulated in aged mice.

Aged mice exhibited increased expression of p47phox and gp91phox, the key subunits of NADPH oxidase, and enhanced reactive oxygen species (ROS) primarily in neuronal cells of the hippocampus (Fig. 1f–i; Supplementary Fig. 1). However, expression of GR in the hippocampus was reduced in aged mice (Fig. 1j). Aged mice lacking p47phox (p47phox KO) did not show increased oxidative stress compared with young mice, although they expressed more gp91phox and p40phox (Fig. 1k–n), suggesting that NADPH oxidase, particularly p47phox, is a crucial factor in ROS accumulation in aged mice.

GC (corticosterone) treatment in HT22 hippocampal neuronal cells increased ROS levels and p47phox expression (Supplementary Fig. 2a-d). Mice exposed to a 2-h restraint for 14 days (RST14d) exhibited increased expression of p47phox, p67phox, and gp91phox in the hippocampus, and showed increased levels of dihydroethidium (DHE)-sensitive ROS and lipid peroxidation in the hippocampus (Fig. 1o–t), whereas ROS accumulation and lipid peroxidation in the hippocampus did not increase in p47phox KO mice after treatment with chronic stress (Fig. 1r–t). These results suggest that NADPH oxidase upregulated by stress is responsible for stress-induced ROS accumulation in the hippocampus.

**Aged mice are highly vulnerable to stress-induced depression.** Given that aging brains exhibited increased expression of NADPH oxidase and oxidative stress in the hippocampus (Fig. 1f–i), similar to changes induced by chronic stress (Fig. 1o–t), we examined whether aged mice are sensitive to stress-induced depression or exhibit depressive behaviors. Young (2-month-old) mice treated with RST14d exhibited reduced sociability in the two-choice social interaction test (SIT), and increased immobility in the forced swim test (FST) (Fig. 2a–f). In contrast, young p47phox KO mice did not show depression-like behaviors after RST14d treatment (Supplementary Fig. 3). Young mice subjected to 2-h restraint for 5 days (RST5d) showed control-like behaviors, whereas aged mice exposed to RST5d exhibited reduced sociability in the SIT, and increased immobility in the FST (Fig. 2a–f). However, in the absence of stress exposure, aged mice were not depressive in each of the behavioral tests (Fig. 2a–f). K-Means cluster analysis (k = 2), an unsupervised machine learning algorithm, of individual animals in the sociability × FST matrix indicated that aged mice subjected to RST5d exhibited depressive-like behaviors comparable to that displayed by young mice after exposure to RST14d (Fig. 2g, h). Further analyses by grouping into three or four clusters (k = 3 and 4, respectively) indicated that more individuals among RST5d-treated aged mice exhibited severe phenotypes compared with RST14d-treated young mice (Supplementary Fig. 4c-f). Together, these results suggest that aged mice are highly vulnerable to stress-induced depression.

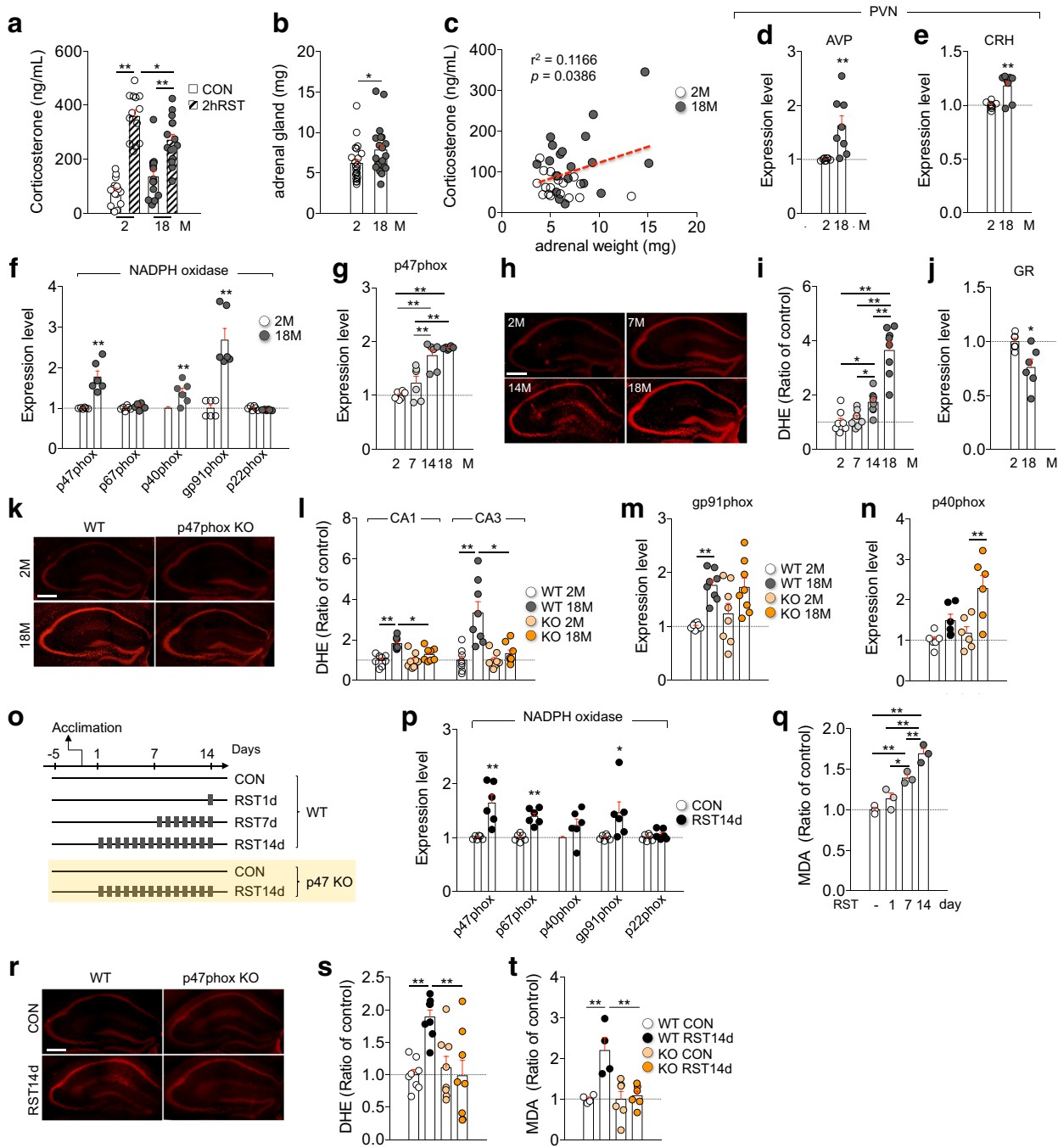

Real-time PCR analysis indicated that basal expression levels of p47phox and gp91phox in aged mice were higher than those in young mice. Furthermore, expression levels of p47phox and gp91phox in aged mice increased further after exposed to RST5d, and their expression levels in aged mice after RST5d exposure were greater than those in young mice after RST14d exposure (Fig. 2i, j).

**p-AMPK regulates stress-induced increase of p47phox.** p47phox expression is regulated by AMPK, p38, Akt, and PI3K[28,40,41]. We found that the expression of Prkaa1, an alpha subunit of AMPK, was reduced in the hippocampus of young mice subjected to RST14d and in the hippocampus of aged mice compared with that of naïve young mouse control (Fig. 3a, b). Western blot analysis indicated that the level of p-AMPK was also

reduced in aged mice relative to young mice, and this decrease was partly due to reduced expression of AMPK (Fig. 3c). Immunofluorescence staining revealed that p-AMPK and p47phox were co-expressed at the single-cell level in pyramidal neurons of the hippocampus, although expression levels of p47phox were negatively correlated with those of p-AMPK (Fig. 3d, e).

GC treatment in HT22 cells decreased p-AMPK levels while increasing p47phox expression and ROS levels (Fig. 3f–h). GC-induced increase of p47phox and ROS levels was blocked by AICAR, an AMPK activator (Fig. 3g, h), whereas treatment with compound C (CC, an AMPK inhibitor) increased p47phox expression and ROS levels (Fig. 3i, j). These results suggest that p-AMPK mediates GC-induced expression of p47phox and ROS accumulation in hippocampal cells (Fig. 3k).

**Fig. 1 Aging- and stress-induced ROS accumulation in the hippocampus was regulated by NADPH oxidase. a** Serum corticosterone levels after 2-h restraint or control of mice at 2 months and 18 months of age ($n = 15$ for 2 M CON, 13 for 2M 2hRST, 14 for 18M CON, and 15 for 18M 2hRST groups; Two-way ANOVA, age, $F_{(1,53)} = 0.6871$, $p = 0.4109$; stress, $F_{(1,53)} = 102.6$, $p < 0.0001$; age x stress, $F_{(1,53)} = 12.23$, $p = 0.001$). **b** Adrenal gland weights ($n = 23$ for 2M and 20 for 18M mouse groups). **c** Relationship between adrenal gland weights and serum corticosterone levels. **d**, **e** Transcript levels of AVP and CRH in the PVN ($n = 8$, each). **f** Transcript levels of NADPH oxidase subunits in the hippocampus ($n = 6$, each). **g** Transcript levels of p47phox in the hippocampus ($n = 6$, each; One-way ANOVA, $F_{(3,20)} = 24.85$, $p < 0.0001$). **h**, **i** DHE-stained hippocampus of mice at the indicated age. Quantification levels ($n = 8$ mice/group; One-way ANOVA, $F_{(3,28)} = 39.1$, $p < 0.0001$). **j** GR transcript level in the hippocampus ($n = 6$, each). **k**, **l** DHE-stained hippocampus of wild-type and p47phox KO mice. Quantification levels ($n = 8$ mice/group; CA1, Two-way ANOVA, age, $F_{(1,28)} = 23.92$, $p < 0.0001$; genotype, $F_{(1,28)} = 5.744$, $p = 0.0235$; age x genotype, $F_{(1,28)} = 5.192$, $p = 0.0305$; CA3, Two-way ANOVA, age, $F_{(1,28)} = 18.31$, $p = 0.0002$; genotype, $F_{(1,28)} = 10.44$, $p = 0.0031$; age x genotype, $F_{(1,28)} = 10.31$, $p = 0.0031$). **m**, **n** gp91phox and p40phox expression levels (gp91phox, $n = 8$, each; Two-way ANOVA, age, $F_{(1,28)} = 20.42$, $p = 0.0001$; genotype, $F_{(1,28)} = 0.5004$, $p = 0.4852$; age x genotype, $F_{(1,28)} = 0.9644$, $p = 0.3345$; p40phox, $n = 6$, each; Two-way ANOVA, age, $F_{(1,20)} = 16.26$, $p = 0.0007$; genotype, $F_{(1,20)} = 6.036$, $p = 0.0233$); age x genotype, $F_{(1,20)} = 2.462$, $p = 0.1323$). **o** Experimental designs. **p** NADPH oxidase subunit expression levels ($n = 6$, each). **q** MDA levels in the hippocampus of the indicated groups ($n = 3$, each; One-way ANOVA, $F_{(3,8)} = 31.2$, $p < 0.0001$). **r**, **s** DHE-reactive ROS levels in the hippocampus. Quantification levels ($n = 8$ mice/group; Two-way ANOVA, genotype, $F_{(1,28)} = 6.016$, $p = 0.0207$; stress, $F_{(1,28)} = 5.581$, $p = 0.0253$; genotype x stress, $F_{(1,28)} = 9.85$, $p = 0.004$). **t** MDA levels in the hippocampus of p47phox KO mice and wild-type mice after treatment with RST14d ($n = 4$, each for wild-type, $n = 6$, each for KO mice; Two-way ANOVA, genotype, $F_{(1,16)} = 8.747$, $p = 0.0093$; stress, $F_{(1,16)} = 11.54$, $p = 0.0037$; genotype x stress, $F_{(1,16)} = 8.53$, $p = 0.01$). Scale bars, 400 μm. *$p < 0.05$ and **$p < 0.01$. One-way ANOVA followed by a Newman–Keuls post hoc test and two-way ANOVA followed by a Bonferroni post hoc test.

Next, we examined whether AMPK regulates stress-induced depressive behaviors. Mice treated with RST14d exhibited reduced levels of p-AMPK and increased expression of p47phox in the hippocampus, whereas AICAR (an AMPK activator) administration during RST14d treatment blocked the reduced p-AMPK levels and increased p47phox and gp91phox expression in the hippocampus. Furthermore, AICAR treatment blocked stress-induced reduced sociability in the SIT and increased immobility in the TST and FST (Supplementary Fig. 5a-f). Consistent with these data, mice treated with CC for 5 days had reduced levels of p-AMPK and increased expression of p47phox and gp91phox in the hippocampus and exhibited depression-like behaviors in the SIT, TST and FST. However, p47phox knockout mice treated with CC did not show depression-like behaviors in those behavioral tests (Supplementary Fig. 5g-l). Together, these results suggest that AMPK regulates stress-induced upregulation of p47phox in the hippocampus, and stress-induced depressive behavior.

**Stress-induced Ppp2ca reduces p-AMPK and promotes depression**. Among the known upstream factors that regulate AMPK, we found that protein phosphatase 2A (Ppp2ca) expression was increased by GC in HT22 cells (Fig. 4a, b). Consistently, GR binding at the *Ppp2ca* promoter increased after GC treatment (Fig. 4c). Moreover, siRNA-mediated Ppp2ca knockdown in HT22 cells increased p-AMPK levels (Fig. 4d). These results suggest that Ppp2ca functions as a negative regulator of AMPK in neuronal cells (Fig. 4e). Indeed, Ppp2ca expression increased in the hippocampus in an age-dependent manner (Fig. 4f), consistent with the decreased levels of p-AMPK in aged mice (Fig. 3c).

Mice subjected to RST14d exhibited increased expression of Ppp2ca in the hippocampus, whereas siRNA-mediated knockdown of Ppp2ca in the hippocampus suppressed stress-induced depression-like behaviors (Fig. 4g–k). Ppp2ca knockdown also reversed stress-induced increased expression of p47phox and gp91phox (Fig. 4l). These results indicate that Ppp2ca functions as an upstream regulator of p-AMPK, p47phox, and gp91phox, and that increased expression of Ppp2ca in the hippocampus promotes depression-like behaviors.

**Aging and stress decrease SUV39H1 expression**. We investigated whether an epigenetic mechanism is involved in aging brains. Real-time PCR analysis indicated that among the epigenetic factors controlling histone acetylation and methylation examined, the transcript levels of the histone deacetylases Hdac2 and Sirt1 were increased and decreased, respectively, in the hippocampus of aged mice, whereas that of the methyltransferase SUV39H1 was decreased (Fig. 5a), and their changes advanced in an age-dependent manner (Fig. 5b–d). SUV39H1 protein was expressed in pyramidal and granule cells in the hippocampus and consistent with the transcript level, its expression levels in the CA1, CA3, and DG regions were decreased in aged mice (Fig. 5e–g).

In HT22 cells, GC treatment decreases SUV39H1 and Hdac2 expression, although the GC-induced change of Hdac2 expression occurs only at high GC concentration[38]. Consistent with the previous report, GC treatment in HT22 cells decreased SUV39H1 expression, whereas the GC-induced change was blocked by siRNA-mediated inhibition of glucocorticoid receptors (GR) (Supplementary Fig. 6a). Regarding the complex expression patterns of Hdac2 in aged mice and in GC-treated HT22 cells, we therefore focused on SUV39H1 in the present study. AMPK-activator (AICAR) treatment in HT22 cells antagonized GC-induced suppression of SUV39H1, whereas AMPK-inhibitor (CC) treatment decreased SUV39H1 expression (Supplementary Fig. 6b, c), suggesting that SUV39H1 expression is regulated by AMPK. A chromatin immunoprecipitation (ChIP) assay indicated that SUV39H1 binding to the promoter of the *p47phox* and *gp91phox* decreased in HT22 cells after GC treatment (Supplementary Fig. 6d, e).

SUV39H1 expression in the hippocampus was downregulated after treatment with RST14d. However, siRNA-mediated Ppp2ca knockdown in the hippocampus (Fig. 5h) or AICAR (an AMPK activator) treatment during RST14d reversed the stress-induced decrease of SUV39H1 (Fig. 5i). Conversely, repeated CC-treatment in normal mice suppressed SUV39H1 expression (Fig. 5j). SUV39H1 expression was reduced after treatment with RST14d, but not RST5d, in young mice. SUV39H1 expression in aged mice was lower than that in young mice, and its expression decreased further after RST5d treatment (Fig. 5k).

Immunofluorescence staining indicated that SUV39H1 and p47phox were co-localized at the single-cell level in pyramidal neurons of the hippocampus, where their expression levels were negatively correlated (Supplementary Fig. 6f, g).

Next, we investigated the mechanism by which p-AMPK regulates p47phox. We found that AMPK activation with AICAR in HT22 cells increased p-CREB level, whereas its inhibition with CC decreased p-CREB (Fig. 5l, m). Furthermore,

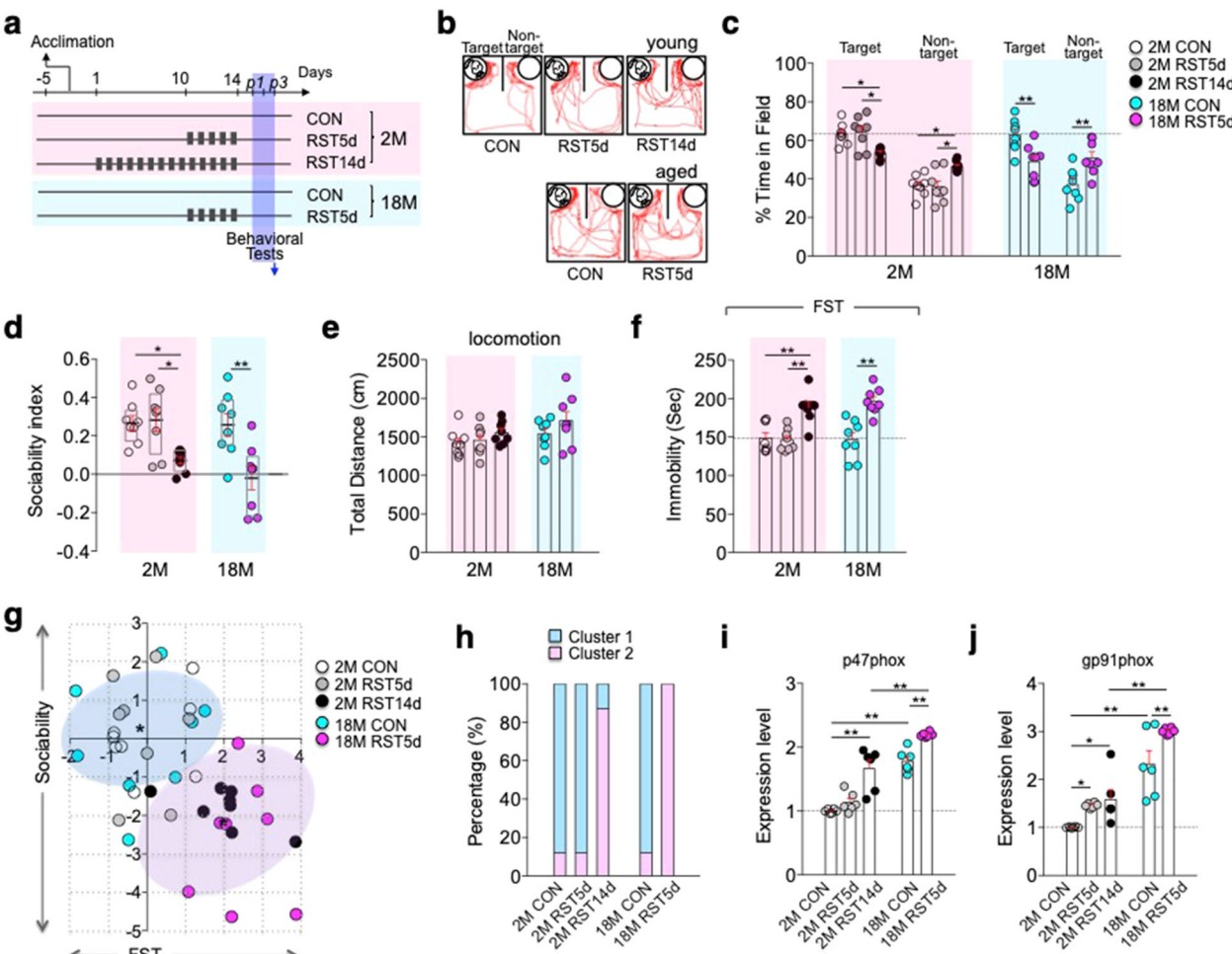

**Fig. 2 Aged mice were highly susceptible to stress-induced depression. a** Experimental design for treatment with daily 2-h restraint for 5 days (RST5d) or 14 days (RST14d) in young (2M) and aged (18M) mice, and following behavioral tests. **b, c** Representative tracking plots, time spent in the target and non-target fields ($n = 8$ mice/group; target field, One-way ANOVA, $F_{(4,35)}=7.416$, $p = 0.0002$; non-target field, One-way ANOVA, $F_{(4,35)}=7.428$, $p = 0.0002$). **d** Sociability index ($n = 8$ mice/group; One-way ANOVA, $F_{(4,35)}=7423$, $p = 0.0002$). **e** Total locomotion in the sociability test ($n = 8$ mice/group; One-way ANOVA, $F_{(4,35)}=2.116$, $p = 0.0996$). **f** Immobility time in the FST ($n = 8$ mice/group; One-way ANOVA, $F_{(4,35)}=12.97$, $p < 0.0001$). **g, h** K-Means clustering of all individuals in the sociability test x FST matrix plotting with z-scores (standard deviations) and proportion of each group in the two clusters (cluster1: 87.5% for 2M CON, 87.5% for 2M RST5d, 12.5% for 2M RST14d, 87.5% for 18M CON, and 0% for 18M RST5d; cluster2: 12.5% for 2M CON, 12.5% for 2M RST5d, 87.5% for 2M RST14d, 12.5% for 18M CON, and 100% for 18M RST5d). **i, j** Expression levels of p47phox and gp91phox transcripts in the hippocampus of indicated groups ($n = 6$, each; p47phox, One-way ANOVA, $F_{(4,25)}=43.24$, $p < 0.0001$; gp91phox, One-way ANOVA, $F_{(4,25)}=25.3$, $p < 0.0001$). *$p < 0.05$ and **$p < 0.01$. One-way ANOVA followed by a Newman–Keuls post hoc test.

siRNA-mediated inhibition of CREB decreased SUV39H1 expression while increasing p47phox expression (Fig. 5n). These results suggest that p-AMPK regulates p47phox expression via p-CREB and SUV39H1 (Fig. 5o).

**SUV39H1 negatively regulates p47phox and gp91phox expression**. Next, we examined whether SUV39H1 regulated the expression of p47phox and gp91phox in vivo. siRNA-mediated knockdown of SUV39H1 in the CA3 of the hippocampus increased expression of p47phox and gp91phox, but not p67phox (Fig. 6a, b). Furthermore, the siRNA-mediated knockdown of SUV39H1 enhanced ROS accumulation (Fig. 6c, d).

ChIP analysis indicated that SUV39H1 binding to the promoter of the *p47phox* and *gp91phox* decreased in the hippocampus of aged mice compared with young mice (Fig. 6e, f, j, k). The levels of tri- and di-methylated histone-3 lysine-9 (H3K9) residue at the promoter of the *p47phox* and *gp91phox* were also consistently reduced in aged mice compared with those in young mice (Fig. 6g, h, l, m). Conversely, the levels of

acetylated H3K9 at the promoter of the *p47phox* and *gp91phox* increased in aged mice compared with those in young mice (Fig. 6i, n). These changes were not observed at the promoter of the *p67phox*, although tri-methylated H3K9 was decreased in aged mice (Supplementary Fig. 7). Together, these results suggest that SUV39H1 negatively regulates the expression of p47phox and gp91phox.

**RA offers anti-depressive effects in young and aged mice**. Rosmarinic acid (RA) activates AMPK[42–44], and upregulates SUV39H1 expression[38]. Therefore, we examined whether RA produces similar effects on the AMPK signing pathway in neuronal cells. RA treatment in HT22 cells reversed GC-induced changes in the expression of Ppp2ca and SUV39H1 (Fig. 7a–c), and GC-induced increased expression of p47phox and accumulation of ROS (Fig. 7d–f). Consistent with these results, RA infusion into CA3 subregion of the hippocampus decreased Ppp2ca, p47phox, and gp91phox expression. Immunohistological analysis indicated that RA infusion into the CA3 subregion

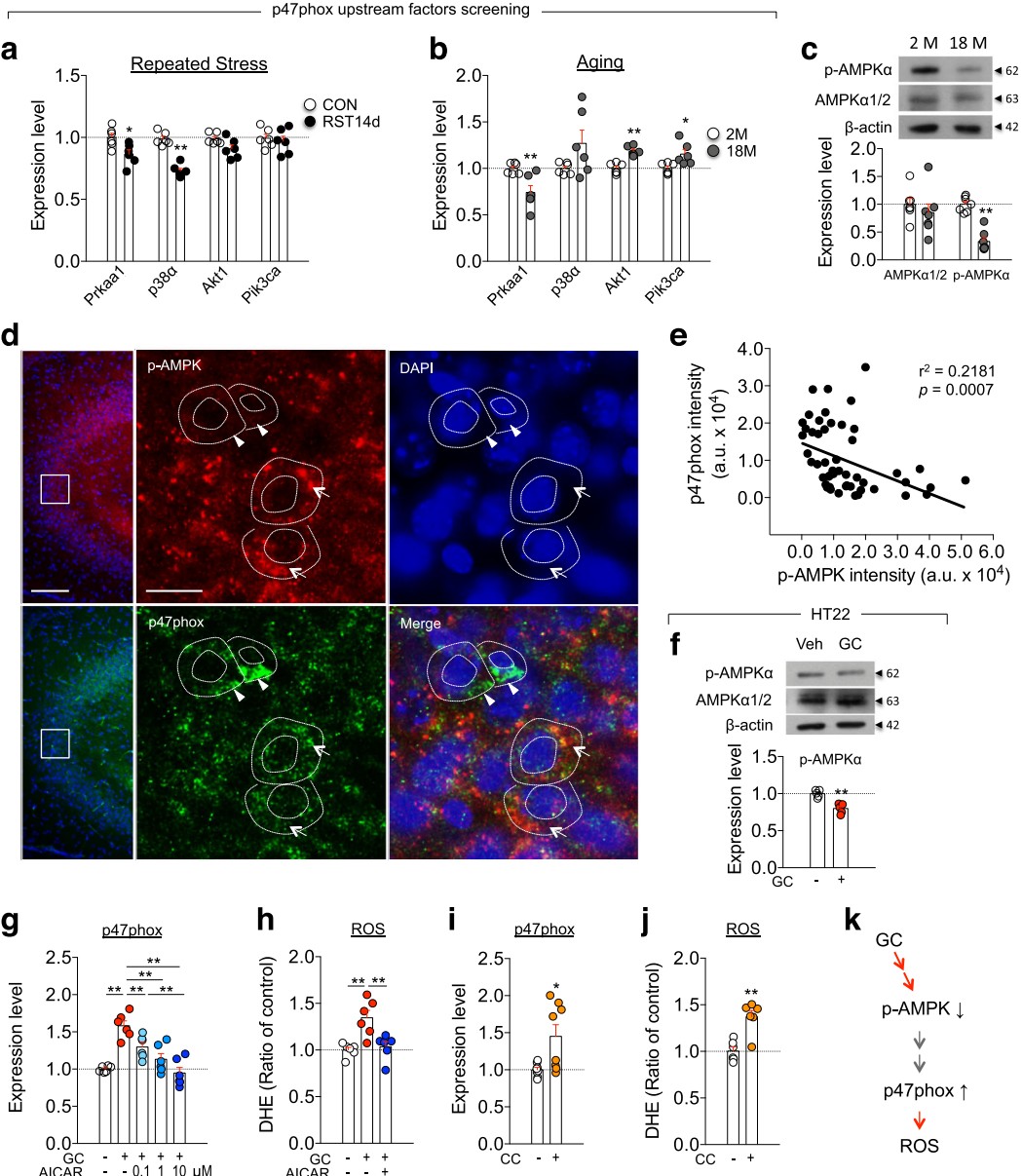

**Fig. 3 Aging- and stress-induced increase of p47phox expression was regulated by AMPK. a**, **b** Expression levels of Prkaa1, p38α, Akt1, and Pik3ca transcripts in the hippocampus of mice treated with RST14d and their control, and of normal young (2M) and aged (18M) mice ($n = 6$, each). **c** Western blot data showing p-AMPK and AMPK levels in the hippocampus of young and aged mice ($n = 7$, each). **d** Photomicrographs showing co-localization of p-AMPK and p47phox in CA3 pyramidal neurons of mice. Scale bars, 100 μm (left), 20 μm (right). **e** Inverse relationship in the expression levels between p-AMPK and p47phox (arrows and arrow heads, respectively) ($n = 49$ cells). **f** Western blot data showing p-AMPK levels in HT22 cells treated with GC (400 ng/ml) for 24 h ($n = 5$, each). **g**, **h** p47phox transcript levels and DHE-reactive ROS levels in HT22 cells treated with GC (400 ng/ml) or GC plus AICAR (p47phox, $n = 6$, each; One-way ANOVA, $F_{(4,25)} = 16.47$, $p < 0.0001$; DHE, $n = 6$, each; One-way ANOVA, $F_{(2,15)} = 10.71$, $p = 0.0013$). **i**, **j** p47phox transcript level and DHE-reactive ROS levels in HT22 cells treated with Compound C (1 μM) (p47phox, $n = 8$, each; DHE, $n = 6$, each). **k** A summary of the signaling pathways of p-AMPK and p47phox. *$p < 0.05$ and **$p < 0.01$. One-way ANOVA followed by a Newman–Keuls post hoc test.

increased expression levels of SUV39H1 and p-AMPK in pyramidal cells, while decreasing p47phox expression (Supplementary Fig. 8). Next, we examined whether systemic administration of RA produced similar effects. RA injection in aged mice during RST5d treatment reversed the stress-induced altered expression of Ppp2ca, SUV39H1, p47phox and gp91phox in the hippocampus (Fig. 7g–k). Furthermore, RA injection in aged mice blocked the stress-induced reduced sociability in the SIT and suppressed the stress-induced increased immobility in the FST (Fig. 7l–p).

RA treatment also reversed the altered expression of Ppp2ca, p-AMPK, SUV39H1, p47phox and gp91phox in the hippocampus of young mice subjected to RST14d treatment (Supplementary Fig. 9a–f). ChIP assay indicated that RA treatment increased SUV39H1 binding to the prompter of the *p47phox* and *gp91phox* (Supplementary Fig. 9g, h). Furthermore, RA treatment in RST14d-treated young mice increased the reduced sociability in the two-choice sociability test and suppressed the increased immobility in the TST and FST (Supplementary Fig. 9i–l). *K*-Means cluster analysis of individual animals in the

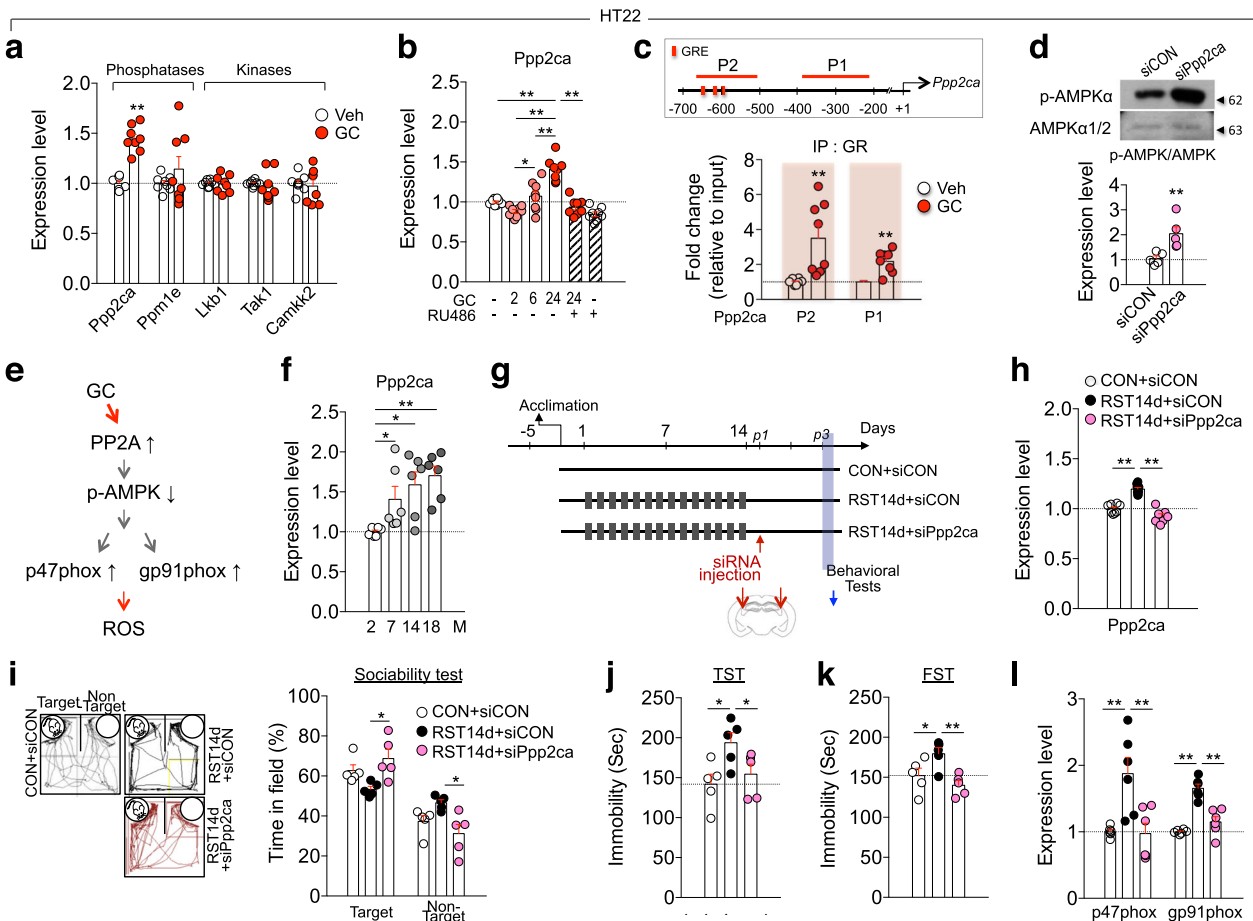

**Fig. 4 Protein phosphatase 2a (Ppp2ca) functioned as an upstream regulator for aging- and stress-induced changes of AMPK-p47phox. a** Expression levels of phosphatases (Ppp2ca and Ppm1e) and kinases (Lkb1, Tak1, and Camkk2) in HT22 cells treated with GC (400 ng/ml) for 24 h ($n = 8$, each). **b** Ppp2ca transcript levels in HT22 cells treated with GC for 2, 6, or 24 h, GC (for 24 h) plus RU486, and RU486 alone ($n = 8$, each; GC2h vs. GC6h vs. GC24h, One-way ANOVA, $F_{(3,28)} = 26.77$, $p < 0.0001$; GC + RU486, Two-way ANOVA, GC, $F_{(1,28)} = 77.75$, $p < 0.0001$; RU486, $F_{(1,28)} = 49.79$, $p < 0.0001$; GC x RU486, $F_{(1,28)} = 20.04$, $p = 0.0001$). **c** ChIP assay data showing GR binding levels to the *Ppp2ca* promoter in HT22 cells treated with for 24 h ($n = 8$, each). Red boxes, GRE. **d** Western blot data showing p-AMPK levels in HT22 cells treated with siPpp2ca and siCON ($n = 5$, each). **e** A summary of the signaling pathway of PP2A, p-AMPK, and p47phox. **f** Ppp2ca transcript levels in the hippocampus of mice at the indicated age ($n = 6$, each; One-way ANOVA, $F_{(3,20)} = 5.697$, $p = 0.0055$). **g** Experimental design. Mice were treated with RST14d, and then received siPpp2ca or siCON injection in CA3 region (red arrows) on post-stress day 1, and were placed on the behavioral tests on post-stress day 3. Arrow, tissue prep point. **h** Ppp2ca transcript levels in the CA3 of mice treated with RST14d and siPpp2ca or siCON injection ($n = 8$, each; One-way ANOVA, $F_{(2,21)} = 52.76$, $p < 0.0001$). **i** Representative tracking plots and time spent in the target and non-target fields in the sociability test ($n = 5$ mice/group; target field, One-way ANOVA, $F_{(2,12)} = 5.973$, $p = 0.0173$; non-target field, One-way ANOVA, $F_{(2,12)} = 5.77$, $p = 0.0175$). **j, k** Immobility time in the TST and FST for mice treated with RST14d and injected with siPpp2ca or siCON ($n = 5$ mice/group; TST, One-way ANOVA, $F_{(2,12)} = 4.616$, $p = 0.0326$; FST, One-way ANOVA, $F_{(2,12)} = 6.817$, $p = 0.0105$). **l** p47phox and gp91phox transcript levels in the CA3 of mice treated with RST14d and siPpp2ca or siCON injection ($n = 6$, each; p47phox, One-way ANOVA, $F_{(2,15)} = 9.357$, $p = 0.0023$; gp91phox, One-way ANOVA, $F_{(2,15)} = 34.95$, $p < 0.0001$). $*p < 0.05$ and $**p < 0.01$. One-way ANOVA followed by a Newman–Keuls post hoc test and two-way ANOVA followed by a Bonferroni post hoc test.

sociability × TST × FST matrix indicated that RA treatment shifted individuals from the cluster containing RST14d to the cluster containing control mice (Supplementary Fig. 9m, n). These results suggest that RA confers anti-depressant-like effects in both young and aged mice subjected to depressive behavior-promoting stress challenge.

## Discussion

Several lines of evidence suggest that aging proceeds with various cellular and physiological changes that are overlapped with those induced by chronic stress in young animals. An important age-dependent change is dysregulation of the HPA axis[13,45,46]. Stress-induced HPA-axis dysfunction is represented by increased basal serum GC levels, reduced response of GC release to stressors, but protracted GC clearance[47–49]. Consistent with those reports, aged

mice had higher basal serum corticosterone levels and impaired HPA-axis response to stressful stimuli compared with young mice (Fig. 1a–c). These results suggest that the stress system in aged mice is not tightly controlled under normal physiological conditions as much as young mice and also do not effectively respond to incoming stress compared with young mice. Aged mice also exhibited increased expression of NADPH oxidase and ROS levels in the hippocampus (Fig. 1f–i). We provided evidence that those physiological and molecular changes in aged mice resemble the respective changes in mice that showed persistent depressive behaviors following chronic stress (Figs. 1, 2). Aged mice exhibited increased oxidative stress (Fig. 1h, i, k, l), increased expression of p47phox and gp91phox (Fig. 1f, g), and reduced expression of SUV39H1 (Fig. 5). However, naïve aged mice were not depressive in the behavioral tests (Fig. 2b–h), suggesting that

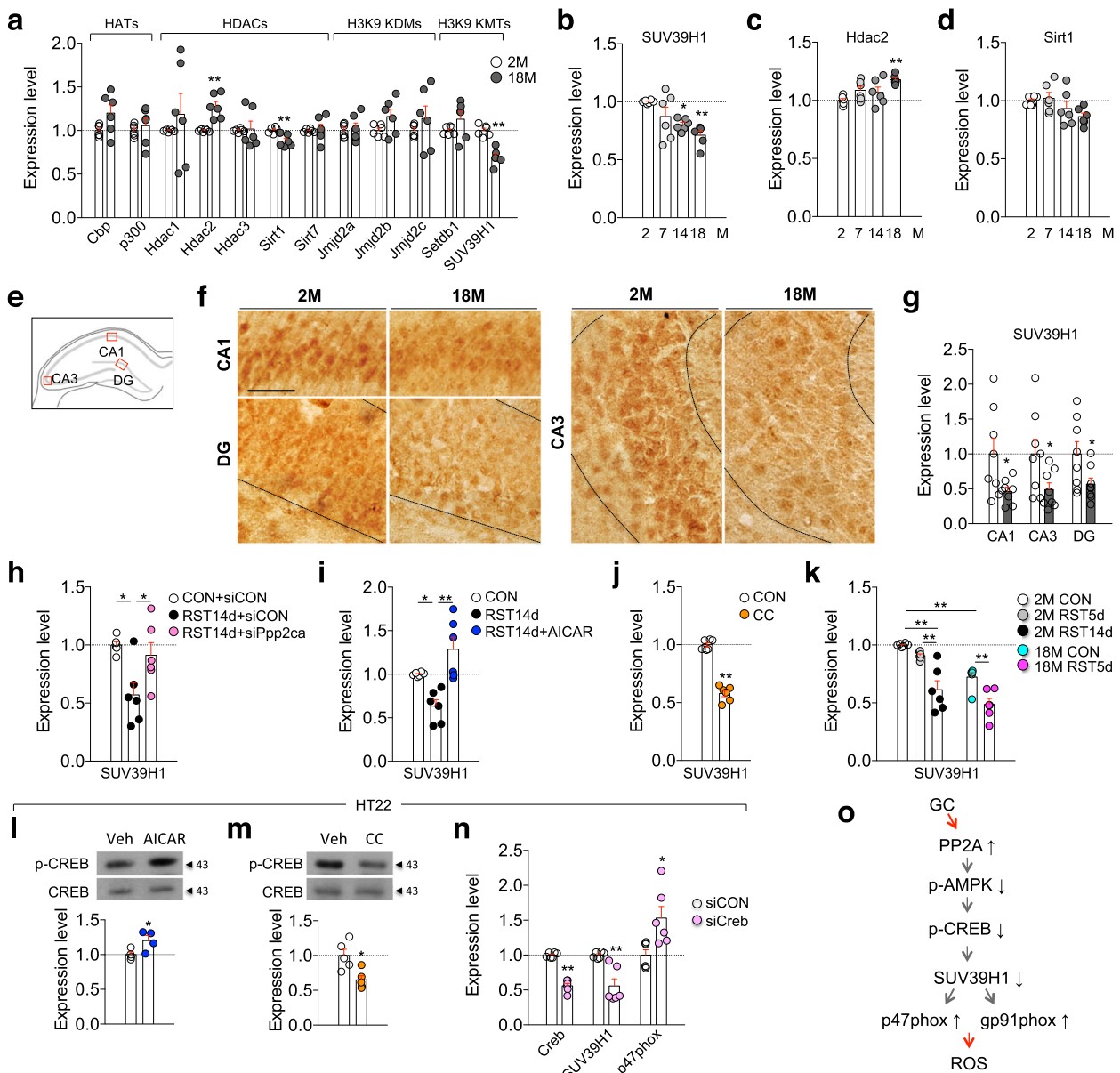

**Fig. 5 SUV39H1 mediated aging- and stress-induced upregulation of p47phox and gp91phox expression. a** Transcript levels of histone acetyltransferases (HATs: Cbp and p300), histone deacetylases (HDACs: Hdac1, Hdac2, Hdac3, Sirt1, and Sirt7), H3K9 lysine-specific demethylases (H3K9 KDMs: Jmjd2a, Jmjd2b, and Jmjd2c) and H3K9 lysine-specific methyltransferases (H3K9 KMTs: Setdb1 and SUV39H1) in the hippocampus of mice at 2 and 18 months of age ($n = 6$, each). **b–d** SUV39H1, Hdac2, and Sirt1 transcript levels in the hippocampus of mice at 2, 7, 14, and 18 months ($n = 6$, each; SUV39H1, One-way ANOVA, $F_{(3,20)}=7.065$, $p = 0.002$; Hdac2, One-way ANOVA, $F_{(3,20)}=5.783$, $p = 0.0051$; Sirt1, One-way ANOVA, $F_{(3,20)}=2.666$, $p = 0.0756$). **e–g** Diagram (**e**) of the hippocampus with the areas (red box) used for the high magnification images (**f**). Scale bars, 50 μm. Quantification levels (**g**) ($n = 8$ mice/group). **h–j** SUV39H1 transcript levels in the hippocampus of mice treated with RST14d plus siCON, or RST14d plus siPpp2ca as depicted in Fig. 4g; RST14d plus Veh, or RST14d plus AICAR as depicted in Supplementary Fig. 5a; and CC as depicted in in Supplementary Fig. 5g ($n = 6$, each; **h** One-way ANOVA, $F_{(2,15)}=6.365$, $p = 0.01$; **i** One-way ANOVA, $F_{(2,15)}=12.84$, $p = 0.0006$). **k** SUV39H1 transcript levels in young mice treated with RST5d or RST14d, and aged mice treated with RST5d, and their control ($n = 6$, each; One-way ANOVA, $F_{(4,25)}=21.4$, $p < 0.0001$). **l**, **m** Western blot data showing p-CREB and CREB levels in the HT22 cells treated with AICAR or CC, respectively (AICAR, $n = 4$, each) (CC, $n = 5$, each). **n** Creb, SUV39H1, and p47phox transcript levels in the HT22 cells treated with siCON or siCreb ($n = 6$, each). **o** A summary of the signaling pathway of PP2A, p-AMPK, p-CREB, SUV39H1, p47phox, and gp91phox. *$p < 0.05$ and **$p < 0.01$. One-way ANOVA followed by a Newman–Keuls post hoc test.

aged mice have acquired the ability not to display depressive-like behaviors, although the detailed mechanism remains unknown.

Nonetheless, aged mice appeared to be in a high-stress state in that they exhibited depression-like behavior when exposed to RST5d, a condition that did not produce depression-like behavior in young mice (Fig. 2b–f). Furthermore, K-Means cluster analysis indicated that the proportion of RST5d-treated aged mice exhibiting severe phenotypes was higher compared than that of

RST14d-treated young mice (Fig. 2g, h; Supplementary Fig. 4). These results suggest that aging greatly increases susceptibility to stress, which can cause depressive-like behaviors.

Aging and chronic stress activate common mechanisms centered on NADPH oxidase at several levels. First, the key subunits of NADPH oxidase p47phox and gp91phox increased in an aging-dependent manner (Fig. 1f, g, m). A similar change in NADPH oxidase expression occurred in young mice exposed to

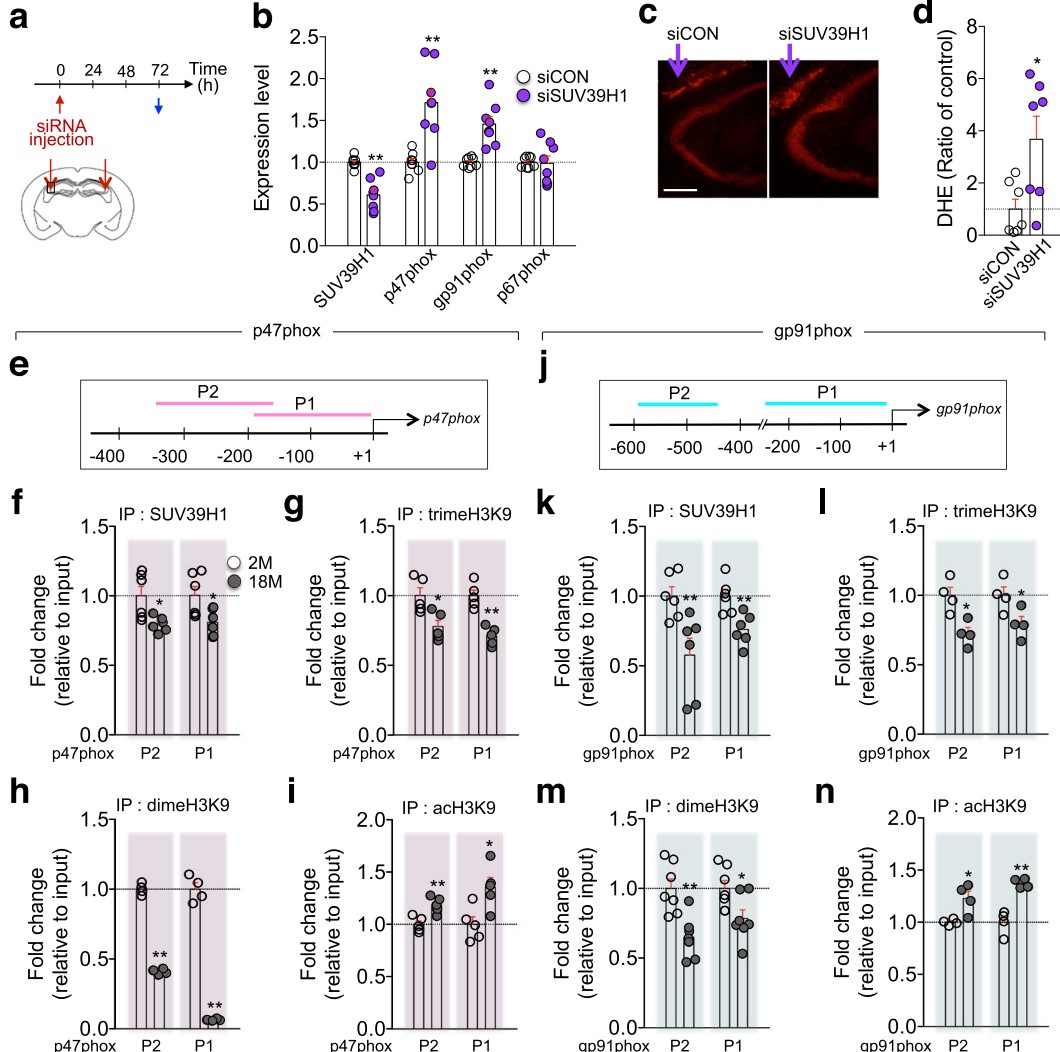

**Fig. 6 SUV39H1 negatively regulated p47phox and gp91phox expression. a** Experimental design. siSUV39H1 or siCON was stereotaxically injected in the CA3 region (red arrows). Blue arrow, tissue preparation point. **b** Expression levels of SUV39H1, p47phox, gp91phox, and p67phox transcripts in the CA3 region ($n = 8$, each). **c** Photomicrographs showing DHE-stained CA3 region of mice injected with siSUV39H1 or siCON. Scale bars, 200 μm. **d** Quantification of DHE-reactive ROS levels ($n = 7$ mice/group). **e** Diagram showing the promoter region of the *p47phox*. **f–i** ChIP-qPCR analysis showing the levels of SUV39H1, trimeH3K9, dimeH3K9, and acH3K9 binding to the promoter of the *p47phox* in the hippocampus of mice at 2 and 18 months of age. P1 and P2, the promoter regions used for ChIP-qPCR analysis. **j** Diagram showing the promoter region of the *gp91phox*. **k–n** ChIP-qPCR analysis showing the levels of SUV39H1, trimeH3K9, dimeH3K9, and acH3K9 binding to the promoter of the *gp91phox* in the hippocampus of mice at 2 and 18 months of age. P1 and P2, the promoter regions used for ChIP-qPCR analysis (p47phox, SUV39H1 $n = 6$, trimeH3K9, $n = 5$, dimeH3K9, $n = 4$, and acH3K9, $n = 5$; gp91phox, SUV39H1 $n = 6$, TrimeH3K9, $n = 4$, DimeH3K9, $n = 7$, and acH3K9, $n = 4$). *$p < 0.05$ and **$p < 0.01$.

chronic stress (Fig. 1p; ref. [20]). Thus, aging and chronic stress commonly upregulate NADPH oxidase in the hippocampus. Second, aged mice exhibited increased ROS accumulation in neuronal cells of the hippocampus, whereas aged p47phox KO mice did not (Fig. 1k, l). Chronic stress increased ROS accumulation in neuronal cells of the hippocampus, whereas chronic stress in p47phox KO mice did not induce such change (Fig. 1r–t). NADPH oxidase is therefore a key factor in aging- and stress-induced ROS accumulation. Third, RA treatment suppressed p47phox and gp91phox expression and protected stress-induced depression (Fig. 7j–n; Supplementary Fig. 9e, f, i-l), while p47phox KO mice were resilient to stress-induced depression. RA treatment reversed aging- and stress-induced changes in upstream factors (eg., Ppp2ca, AMPK and SUV39H1) that regulate p47phox and gp91phox. Together, these results suggest that NADPH oxidase is a critical factor mediating aging-and stress-induced ROS accumulation in the hippocampus and that aging

increases sensitivity to stress due to increased expression of NADPH oxidase.

The epigenetic factor SUV39H1 is a histone methyltransferase that adds methyl groups to H3K9[50], thereby acting as a transcription repressor[38,51]. Both aging and stress downregulated SUV39H1 (Fig. 5), which caused sustained upregulation of p47phox and gp91phox. The deacetylase Sirt1 was also downregulated in the hippocampus of aged mice (Fig. 5a, d), and in the hippocampus of a stress model of depression[52]. Sirt1 regulates the activity of SUV39H1 by deacetylation[53,54]. Furthermore, Sirt1 is regulated by AMPK[55]. Therefore, it will be worth to test the possibility that Sirt1 plays a role in the regulation of p47phox and gp91phox. Aging- and stress-dependent decrease of SUV39H1 was regulated by signaling factors centered on the Ppp2ca-AMPK-CREB signaling pathway (Fig. 5o). Both aging and chronic stress increased the expression of Ppp2ca, whereas siRNA-mediated knockdown of Ppp2ca in the hippocampus

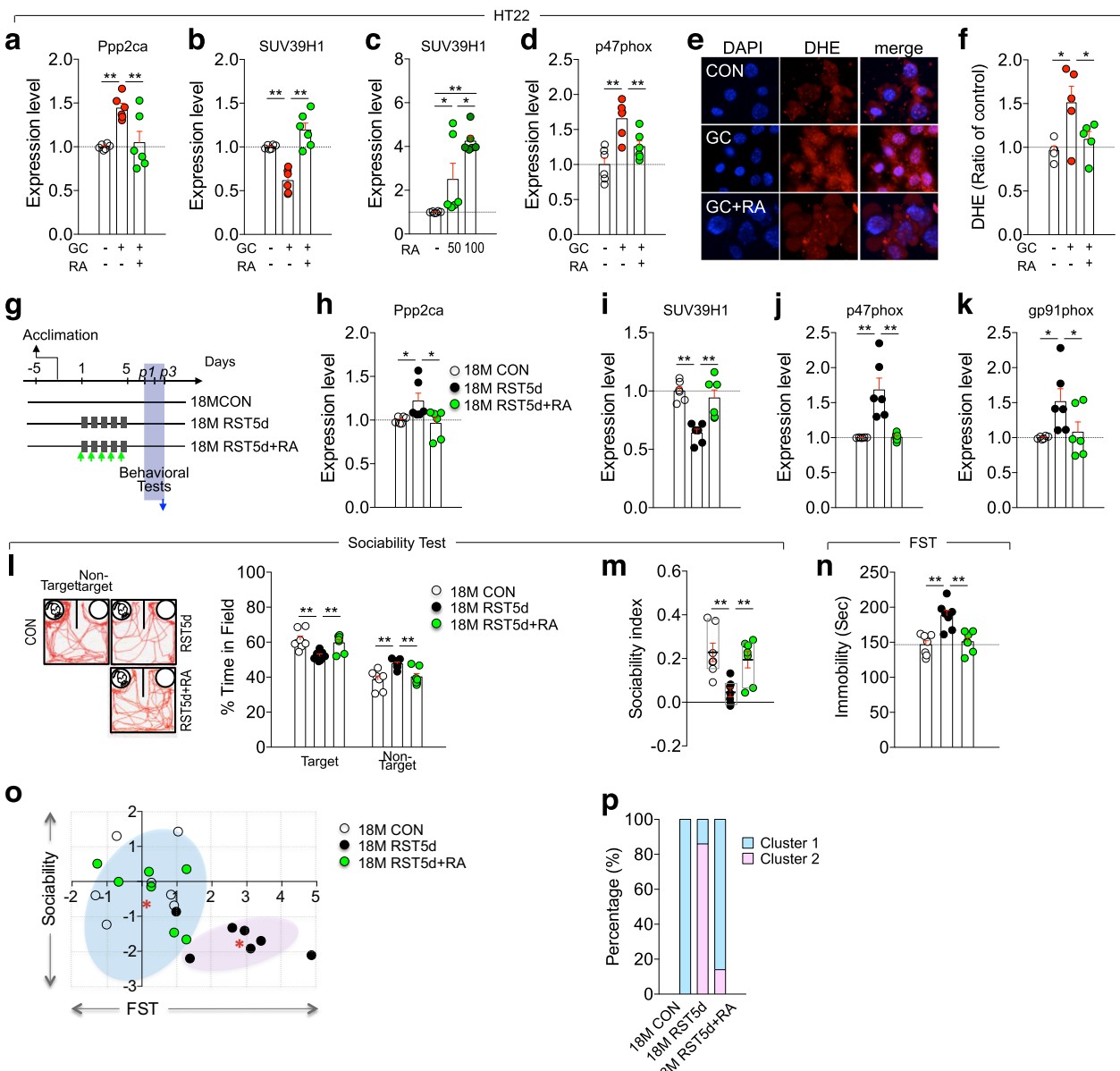

**Fig. 7 RA blocked GC-induced changes in the Ppp2ca-SUV39H1-p47phox pathway and stress-induced depression-like behavior in aged mice. a–d** Ppp2ca, SUV39H1, and p47phox transcript levels in HT22 cells treated with GC GC + RA, or RA alone. RA, 50 or 100 μM (n = 6, each; **a** One-way ANOVA, F(2,15)=8.933, p = 0.0028; **b** One-way ANOVA, F(2,15)=27.62, p < 0.0001; **c** One-way ANOVA, F(2,15)=13.22, p = 0.0005; **d** One-way ANOVA, F (2,15)=12.11, p = 0.0007). **e**, **f** Photomicrographs showing DHE-reactive ROS levels in HT22 cells treated with GC, and GC plus RA, and quantification levels (n = 5, each; One-way ANOVA, F(2,12)=5.31, p = 0.0223). **g** Experimental design for treatment with daily 2-h restraint for 5 days (RST5d) or RST5d plus RA in aged (18M) mice, and following behavioral tests. Arrow, time point for tissue preparation. **h–k** Expression levels of Ppp2ca, SUV39H1, p47phox, and gp91phox transcripts in the hippocampus of indicated groups (n = 6, each; Ppp2ca, One-way ANOVA, F(2,15)=4.458, p = 0.0302; SUV39H1, One-way ANOVA, F(2,15)=15.42, p = 0.0002; p47phox, One-way ANOVA, F(2,15)=15.83, p = 0.0002; gp91phox, One-way ANOVA, F(2,15)=4.249, p = 0.0345). **l** Representative tracking plots and time spent in the target and non-target fields (n = 7 mice/group; target field, One-way ANOVA, F(2,18)=8.003, p = 0.0033; non-target field, One-way ANOVA, F(2,18)=7.969, p = 0.0033). **m** Sociability index in the sociability test (n = 7 mice/group; One-way ANOVA, F(2,18)=7.983, p = 0.0033). **n** Immobility time in the FST (n = 7 mice/group; One-way ANOVA, F(2,18)=13.22, p = 0.0033). **o**, **p** K-Means clustering of individuals in the sociability x FST matrix plotting with z-scores and % composition of each group in the clusters (cluster1, 100% for CON, 14.3% for RST5d, and 85.7% for RST5d + RA; cluster2, 0% for CON, 85.7% for RST5d, and 14.3% for RST5d + RA). *p < 0.05 and **p < 0.01. One-way ANOVA followed by a Newman–Keuls post hoc test.

blocked stress-induced downregulation of SUV39H1 (Fig. 5h). While Ppp2ca negatively regulated AMPK (Fig. 4d), AMPK activation increased p-CREB and SUV39H1, and its inhibition decreased p-CREB and SUV39H1 (Fig. 5i, j, l, m). Together, our results suggest that the Ppp2ca-AMPK-CREB-SUV39H1 pathway functions as a signaling module that regulates aging- and

stress-dependent upregulation of NADPH oxidase and ROS accumulation (Fig. 5o).

SUV39H1 expression can be regulated by the Mkp-1-MAPK/PPARγ pathway, independently of the PP2A-AMPK pathway, in a stress-induced model of depression[38]. Thus, SUV39H1 expression is regulated by Ppp2ca-AMPK and

Mkp-1-MAPK/PPARγ signaling pathways. More interestingly, the expression of the Mkp-1 and Ppp2ca genes are regulated by GC (Fig. 4a, b; [38]). In fact, the Ppp2ca promoter contains three putative glucocorticoid responsive elements (GREs) at −590 to −650 bp (Fig. 4c), and the Mkp-1 promoter also carries a potential GRE at −710 region[38]. To our knowledge, this is the first report describing the functional significance of the putative GREs of the Ppp2ca and Mkp-1 gene promoters. ChIP-qPCR assay indicated GR binding to the promoter regions containing the GREs of the Ppp2ca (Fig. 4c) and Mkp-1 gene[38]. Therefore, age-dependent increase of basal GC levels likely facilitates to enhance expression of the Ppp2ca and Mkp-1 genes. These results raise the following interrelated issues. First, PP2A and Mkp-1 are members of the serine/threonine phosphatase family, which negatively regulate AMPK and MAPK (e.g., ERK1/2 and p38 MAPKs), respectively. Thus, Ppp2ca and Mkp-1 have distinctive cellular targets acting in stress responses. It is possible that the Mkp-1-MAPK/PPARγ pathway regulates p47phox expression, which might contribute to the deviation from the negative correlation between AMPK and p47phox at certain cells (Fig. 3d, e), although direct evidence is not available. Second, despite this divergence, GC-induced upregulation of Ppp2ca and Mkp-1 is converged to commonly downregulate SUV39H1. Increased Mkp-1 suppressed SUV39H1 via MAPK and PPARγ[38], whereas increased Ppp2ca downregulated SUV39H1 via AMPK (Figs. 4, 5). Both the Mkp-1-PPARγ pathway and the PP2A-AMPK pathway commonly regulate SUV39H1 via p-CREB (Fig. 5l, m; ref. [38]). Third, although we demonstrated that the Mkp-1-PPARγ-SUV39H1 pathway functions as a stress-adaptation system in stress-induced depression, the executives of altered function of the Mkp-1-PPARγ-SUV39H1 pathway in stress-induced depression is unknown. Considering that the Ppp2ca-AMPK-CREB-SUV39H1 pathway functions as a signaling module regulating NADPH oxidase, a test of whether NADPH oxidase is a downstream target of the Mkp-1-PPARγ-SUV39H1 pathway in stress-induced changes would also be useful.

Rosmarinic acid (RA) is a natural compound that produces antioxidant and anti-inflammatory effects[56,57]. Independently of its antioxidant property, RA has an ability to suppress GR-dependent increase of Mkp-1 expression, and thereby upregulates SUV39H1 expression via PPARγ[38]. Similar to the inhibition of Mkp-1, RA suppressed GR-dependent increase of Ppp2ca expression, which led to upregulate SUV39H1 (Fig. 7; Supplementary Fig. 9). It is possible that the suppression of Ppp2ca upregulation by RA is related to the inhibition of GR translocation to the nucleus[38]. RA increases p-AMPK levels in colorectal cancer cells and skeletal muscle cells[43,58]. RA increased p-AMPK levels in the brain of mice (Supplementary Fig. 8f, i; Supplementary Fig. 9c). RA increased SUV39H1 binding to the promoters of the p47phox and gp91phox, and thereby suppressed p47phox and gp91phox expression (Supplementary Fig. 9e, h). These results suggest that RA antagonizes stress-induced decrease of AMPK and SUV39H1 and thereby suppresses NADPH oxidase. Furthermore, RA produced anti-depressive effects in aged mice exposed to RST5d (Fig. 7l–n) and young mice subjected to chronic stress (Supplementary Fig. 9i-l). These results suggest that RA could be an effective treatment for stress-induced depression.

The bioavailability of orally fed RA is low. Only 0.9–1% of RA given orally was absorbed into the blood[59,60]. Orally fed RA is hardly delivered to the brain[60]. When RA was intraperitoneally administered in rats, 0.4% of the dose administered was detected in the brain 30 min after injection[60]. The majority of intraperitoneally injected drugs is absorbed through the veins of the mesentery, after being gathered into the portal vein of the liver, and then enters the systemic circulatory system. Therefore, a large proportion of intraperitoneally injected RA could be metabolized or modified in the liver and in the blood. Nonetheless, RA injected intraperitoneally in mice suppressed stress-induced increase of Ppp2ca, upregulated reduced levels of p-AMPK and SUV39H1, and reduced increased expression of p47phox and gp91phox (Supplementary Fig. 9). Those RA-induced signaling responses in the hippocampus of mice (Fig. 7g–k; Supplementary Fig. 9a-h) mirrored those in RA-treated HT22 cells (Fig. 7a–f). Furthermore, RA injection intraperitoneally in mice produced anti-depressive effects in young and aged mice (Fig. 7; Supplementary Fig. 9). Consistent with these results, direct RA infusion into CA3 region via a preimplanted cannula suppressed Ppp2ca expression, increased SUV39H1 and p-AMPK levels, and decreased p47phox and gp91phox levels (Supplementary Fig. 8). Therefore, our results suggest that RA produces anti-depressive effects through upregulation of AMPK and SUV39H1 and suppression of NADPH oxidase, although it is necessary to investigate further, and improve the ways of, the brain availability of peripherally administered RA.

## Materials and methods

**Animals**. Seven-week-old male C57BL/6 mice were purchased from Daehan Bio-Link Inc. (Eumsung, Chungbuk, Korea). p47phox knockout mice[61] were obtained from the Jackson Laboratory. They were backcrossed to C57BL/6 J for more than 10 generations. For genotyping, genomic PCR was performed using the primer set; 5′-TGGAAGAAGCTGAGAGTTGAGG-3′ and 5′-TCCAGGAGCTTATGAAT-GACC-3. When the PCR products were digested with MspI, the 160-base pair (bp) fragment was detected for wild types, and the 102- and 58-bp fragments were for homozygotes. p47phox knockout mice and their wild-type littermates were used in experiments at the indicated age.

Mice were housed in pairs in a standard clear plastic cage filled with chopped wood particles (TAPVEI, Paekna, Estonia) in a temperature- and humidity-controlled environment at 23 °C and 50–60%, respectively, following a 12-h light/dark cycle from 7:00 a.m. to 7:00 p.m. Mice were allowed to free access to food in their cages. All mice were acclimated to the new experimental room for 5 days prior to the experiments. Animals were handled in accordance with the animal-care guidelines of Ewha Womans University (IACUC 16–018).

**Restraint and drugs administration**. Restraint was applied to mice as previously described[62,63]. Mice were individually placed into a 50-ml polypropylene conical tube with multiple holes for ventilation and restrained to prevent back-and forth movement. Restraint was applied for 2-h per day for the number of days indicated. After each daily session of restraint, the mice were returned to their home cages.

Mice were intraperitoneally injected with RA (30 mg/kg) or AICAR (500 mg/kg) 30 min prior to the start of each 2-h restraint session in the co-treatment paradigm. Compound C (10 mg/kg) was injected to normal mice for the indicated days. Rosmarinic acid, AICAR and Compound C were purchased from Tocris (Bristol, UK).

RA infusion into hippocampus subregion was carried out using cannulation as previously described[38,64]. Mice were anesthetized with a mixture (3.5: 1) of ketamine hydrochloride (50 mg/ml) and xylazine hydrochloride (23.3 mg/ml) at a dose of 2.5 μl/g body weight during the implantation of a 26-gauge guide cannula (C315G/SPC, Plastics One, Bilaney, UK). The guide cannula was secured by a dummy cannula (C315DC/SPC, Plastics One). After 7 recovery days, rosmarinic acid or saline was infused into the CA3 region (stereotaxic coordinate: AP, −1.9; ML, ± 2.1; DV,−2.1 mm) through a 33-gauge internal cannula (C315I/SPC, Plastics One) inserted into the guide cannula during anesthetizing with 1.2% isoflurane. Rosmarinic acid (72 μg/ml or 200 μM) was infused into the right side of the CA3 region in a volume of 1.8 μl via a preimplanted cannula. Each animal was infused every 12 h for 2.5 days (total 5 times) and sacrificed after 3 h of the last injection.

**Cell culture and drugs treatment**. HT22 mouse hippocampal cells were cultured as previously described[20,38]. HT22 cells were cultured in Dulbecco's Modified Eagle Medium (DMEM; LM-001–05; Welgene, Gyeongsan, Korea) supplemented with 10% heat-inactivated fetal bovine serum (FBS; FB02–500; Serum Source, NC, USA) and antibiotics (penicillin and streptomycin; LS-202–02; Welgene, Gyeongsan, Korea) at 37ºC in a humidified incubator with 95% air and 5% CO₂.

Cells grown to 70–80% confluence in a 6-well plate or a 100-mm dish were treated with corticosterone, or drugs in DMEM containing 1% FBS. After 24 h, cells were washed in 1X phosphate buffered saline (PBS) and harvested. Corticosterone, RU486, AICAR, or Compound C were applied at the dose indicated. Corticosterone and RU486 were purchased from Tocris.

Cells were transfected with siRNA using Lipofectamine (Invitrogen, Carlsbad, CA, USA) as an reagent in DMEM containing 1% FBS according to the manufacturer's protocol. The final concentration of siRNA was 100 nM, and the concentration of Lipofectamine-2000 was 7.5 μl/well.

The siRNAs used were: siRNA-control (SN-1012), siRNA-GR (#1393304, NM_008173.3), siRNA-Ppp2ca (#1411897, NM_019411.4), and siRNA-Creb (#1342686, NM_009952.2), which were obtained from Bioneer Co. (Daejeon, Korea).

**Measurement of corticosterone levels**. Corticosterone levels were measured as previously described[24]. In brief, immediately after 2-h of restraint, mice were anesthetized with 2.5% avertin (intraperitoneal injection) at a dose of 20 μg/g body weight and blood samples were collected. Control mice were prepared in parallel. Blood samples were collected between 11:00 a.m. and 2:00 p.m. Collected blood samples were centrifuged at 3000 rpm and 4 °C for 15 min and the supernatant was used to measure corticosterone levels. The assays were performed using an ELISA kit (ADI-901-097; Enzo Life Science, Farmingdale, NY, USA) according to the manufacturer's recommendations. Each sample was added to a steroid displacement reagent to release corticosterone from corticosterone-binding globulin, and diluted with an ELSA assay buffer from the ELISA kit. Each blood sample and standard samples were placed onto an anti-sheep IgG-coated plate, and then alkaline phosphatase and sheep polyclonal antibody were added. After 2 h, each well was washed 3 times with a wash solution supplemented from the ELISA kit. Then, p-nitrophenyl phosphatate solution was added and incubated for 1 h. The reaction was stopped with a stop solution, and the plate was read immediately at 405 nm.

**ROS detection and visualization in vitro and in vivo**. Superoxide levels were measured using dihydroethidium (DHE), which is a cell membrane-permeable superoxide-sensitive fluorescent dye (Sigma-Aldrich, St. Louis, MO, USA), as previously described[20]. HT22 cells were seeded at $5 \times 10^3$ cells in 200 μl per well in 96-well plates and incubated overnight. For detection of corticosterone-induced ROS, cells were treated for 24 h with corticosterone or RU486 in DMEM containing 1% FBS. After washing twice with 1X PBS, cells were incubated in DMEM containing 1% FBS and 10 μM DHE for 30 min at 37 °C. After washing twice with 1X PBS, fluorescence levels were measured at 540 nm excitation and 590 nm emission using a spectrofluorometer (SpectraMax i3x, Molecular Devices, Sunnyvale, CA, USA). To visualize ROS in HT22 cells, the cells were cultured on coated coverslips in 24-well plates. The cells were treated as above, and stained cells on a coverslip were mounted on a glass slide with a DAPI-containing mounting solution (H-1200; Vector Lab). Fluorescence images were analyzed using an Olympus BX 51 microscope equipped with a DP71 camera and MetaMorph Microscopy Automation & Image Analysis software (Molecular Device).

For detection of superoxide levels in hippocampal sections, perfused sections were incubated with 1 μM DHE in 1X PBS at room temperature for 5 min while protecting the incubating sections from light using aluminum foil. Sections were washed 3 times with 1X PBS. After placing on a glass slide, the sections were covered with a DAPI-containing mounting solution and coverslipped. Fluorescence images were analyzed using an Olympus BX 51 microscope equipped with a DP71 camera and MetaMorph system.

**Measurement of lipid peroxidation levels**. Lipid peroxidation levels in the brain were analyzed as previously described[20]. Malondialdehyde (MDA) levels were measured using a Bioxytech MDA-586 kit (21044; Oxis Research, Foster City, CA, USA). Briefly, brain tissues were homogenized in 4 volumes of homogenate buffer (1% butylated hydroxytoluene in 1X PBS, pH7.4). Homogenates were centrifuged at 3000 g for 10 min at 4 °C and the supernatant was used for each assay. For each reaction, 10 μl of probucol and 640 μl of diluted R1 reagent (1:3 of methanol/N-methyl-2-phenylindole) were added and then mixed with 150 μl of 12 N HCl. Each reaction was incubated at 45 °C for 60 min to allow the MDA in the sample to react with thiobarbituric acid (TBA). After centrifugation at 10,000 g for 10 min, the supernatant was collected and used to measure the MDA-TBA adduct level at 586 nm. MDA data were normalized over the protein concentration and expressed as a percentage of the sham control value. Protein-bound MDA content was obtained by subtraction.

**Immunohistochemistry**. Immunohistochemical analyses were performed as previously described[38]. Mice were perfused with 4% paraformaldehyde by a transcardiac method. After surgery, the brains were further post-fixed in the same fixative overnight at 4 °C. Brains were coronally sliced into 40-μm sections using a vibratome (Leica VT 1000 S; Leica Instruments, Nussloch, Germany). Free-floating sections were blocked against non-specific binding with 5% bovine serum albumin (BSA) at room temperature for 1 h, and then the sections were reacted with a primary antibody in 5% BSA solution overnight at 4 °C. The sections were washed 3 times with 1X PBS containing 0.1% Tween 20 and incubated for 1 h with secondary antibodies. For immunofluorescence staining, secondary antibody tagged with Dylight 594 (DI-1094; RRID:AB_2336414, DI-2594; RRID:AB_2336412, Vector Lab., Burlingame, CA, USA), or fluorescein isothiocyanate (FITC; sc-2024; RRID:AB_631727, Santa Cruz, TX, USA) was used.

To examine whether ROS was accumulated in neuronal cells, brain sections were incubated with anti-NeuN followed by secondary antibody tagged with Dylight 594. After washed three times in PBS, the brain sections were incubated with 1 μM DHE in 1X PBS at room temperature for 5 min while protecting the incubating sections from light.

After washing, the sections were mounted on a gelatin-coated glass slide with a DAPI-contained mounting solution. Stained sections were analyzed using an Olympus BX 51 microscope equipped with a DP71 camera and MetaMorph Microscopy Automation & Image Analysis software.

To quantify expression relationship between p-AMPK and p47phox and between SUV39H1 and p47phox, we randomly selected neurons in CA3 subregion and read signal intensity levels of p-AMPK and p47phox, and of SUV39H1 and p47phox, at individual cells using a MetaMorph image analysis program. The intensity values of p-AMPK and p47phox expression levels at individual cells were plotted.

The primary antibodies used were: NeuN (1:1000, MAB377; RRID: AB_2298772, Millipore), p-AMPKα (1:100, 2535 s; RRID:AB_331250, Cell Signaling Danvers, USA), p47phox (1:1000, sc-7660; RRID:AB_2298320, Santa Cruz), and SUV39H1 (1:100, GTX112263; RRID:AB_1952113, GeneTex, Irvine, CA, USA).

**Western blot analyses**. Western blot analysis was carried out as previously described[63]. Hippocampal tissue or cultured cells were homogenized in a homogenization buffer (50 mM Tris-HCl, pH 8.0, 150 mM NaCl, 1% Nonidet P-40, 0.1% SDS, and 0.1% sodium deoxycholate) containing a cocktail of protease inhibitors (Roche, Mannheim, Germany) by sonication on ice using an Epishear probe sonicator (Active Motif, Carlsbad, CA, USA) at a 40% power outlet with two rounds of 10-s pulses. The homogenate was centrifuged at 13,000 g and 4 °C for 10 min, and the supernatant was collected. Protein concentration was determined by the Bradford method (Bio-Rad Laboratories. Hercules, CA, USA). The homogenate was mixed with a 6X sample loading buffer and boiled for 5 min. The 20 μg of proteins were resolved on each lane of SDS-PAGE gel, and resolved samples were transferred onto a polyvinylidene fluoride (PVDF) membrane (Bio-Rad Laboratories). Blots were incubated with blocking solution containing 1% BSA in TBST (150 mM NaCl, 50 mM Tris-Cl buffer, pH 7.4, 0.1% Tween 20) followed by incubation with a primary antibody in blocking solution for 3 h at RT. After washing three times with TBST, blots were incubated with secondary antibody in TBST at room temperature for 1 h. Immunoblots were visualized using the reagents of a PicoEPD Western Reagent Kit (EBP-1073; ELPis Biotech, Daejeon, Korea). Western blot images were quantified using Image-Pro Premier 6.0 (MediaCybernetics, MD, USA).

The primary antibodies used were: p-AMPKα (1:1000, 2535s; RRID: AB_331250, Cell Signaling), AMPKα1/2 (1:200, sc-74461; RRID:AB_1118940, Santa Cruz), and p-CREB (1:1000, 06–519, RRID:AB_310153, Millipore), CREB (1:1000, sc-186, RRID:AB_2086021, Santa Cruz), and β-actin (1:2000, sc-47778, RRID:AB_626632, Santa Cruz).

**Quantitative real-time PCR**. Quantitative real-time PCR (qPCR) was performed as previously described[38,63]. Hippocampal tissues were homogenized using a pellet pestle (Kimble, Rockwood, TN, USA), and total RNA from the tissue homogenate was purified with TRIzol reagent (15596-018; Invitrogen). Total RNA from cultured cells was isolated with TRIzol reagent after homogenizing cells by pipetting. Isolated RNA was dissolved in diethyl pyrocarbonate (DEPC)-containing water. RNA concentration was measured with NanoDrop (Thermo Fisher Scientific, Waltham, MA, USA). To prevent genomic DNA contamination, isolated RNA was treated with DNase I (Promega, Madison, WI, USA), and then 1 μg of total RNA was converted to cDNA using a reverse transcriptase system (LeGene Bioscience #6100; San Diego, CA, USA). Real-time PCR was carried out with 4 μl of cDNA (1/8 dilution of the converted cDNA), 10 μl of 2X iQ™ SYBR® Green Supermix (Bio-Rad Laboratories), and 1 μl each of 5 pmol/μl forward and reverse primers in a volume of 20 μl using the CFX 96 Real-Time PCR System Detector (Bio-Rad Laboratories).

Real-time PCR was carried out using the following primers: 5′- ACAGAGTC ATCCCACACCTC-3′ and 5′-GTGGGCAGTTTCAGGTCATC-3′ for p47phox; 5′-CATGAAGCACACCATCCAGTCC-3′ and 5′-GATTTCTCTTCCTGTTTGTCC ATCTG-3′ for p67phox; 5′-GACATCGAGGAGAAGAGGGG-3′ and 5′-GTGAA AGGGCTGTTCTTGCT-3′ for p40phox; 5′- AAGACTCTGTATGGACGGCC-3′ and 5′-GCCGGATTCTGAGTTGGAGA-3′ for gp91phox; 5′-ACAACTGGACA GGAACCTCA-3′ and 5′-TCACCGATGTCAGAGAGAC-3′ for p22phox; 5′-GGGACTGCTACTCCACAGA-3′ and 5′-CCTGGTCTTGGAGCTACGTC-3′ for Prkaa1; 5′-ATTCTGGATTTTGGGCTGGC-3′ and 5′-GTTCTTCCGGTCAAC AGCTC-3′ for p38α; 5′-CACCTTTATCATCCGCTGCC-3′ and 5′-ACACCTC CATCTCTTCAGCC-3′ for Akt1; 5′-TGAAAGTGTGTGGCTGTGAC-3′ and 5′-TTCATGTAGGGTGTGGCTGT-3′ for Pik3ca; 5′- TCTCCGAGTCCCAGGTC AA-3′ and 5′-CCACATAGTCTCCCATAAACAGG-3′ for Ppp2ca; 5′-TGTTT CCCCATGATCCTGCT −3′ and 5′-TCCTCTCTGTCTGGCTTTGTG-3′ for Ppm1e; 5′-GACAGATTAGGCAGCACAGC-3′ and 5-TCTTCCTCCCTCCTCCT CCTC-3′ for Lkb1; 5′-TAAAGGGAGTGCTGCTTGGA-3′ and 5′-TCCTTAGA CCAACAGCGTGT-3′ for Tak1; 5′-GTGGTGAAGCTGGTAGAGGT-3′ and 5′-TGGAAGGTTTGATGTCCCGA-3′ for Camkk2; 5′- GGTTGCCTATGCT AA GAAAGT-3′ and 5′-GATGCCTTGCTTATGTAAACG-3′ for Cbp; 5′-GAC TGAGCAGCGATAATG-3′ and 5′-CAAGGTGTCTCTAGTGTATG-3′ for p300; 5′-CAGTGTGGCTCAGATTCCCT-3′ and 5′- GGGCAGCTCATTA GGGAT

CT-3′ for Hdac1; 5′-GGGACAGGCTTGGTTGTTTC-3′ and 5′-GAGCATCAG CAATGGCAAGT-3′ for Hdac2; 5′-AGAGAGGTCCCGAGGAGAAC-3′ and 5′-ACTCTTGGGGACACAGCATC-3′ for Hdac3; 5′-GATCCTTCAGTGTCATGG TTC-3′ and 5′-ATGGCAAGTGGCTCATCA-3′ for Sirt1; 5′- GAGAGCGAGGA TCTGGTGAC-3′ and 5′-CGTGTAGACAACCAAGTGCC-3′ for Sirt7; 5′-GTCTG GCCTCTTCACTCAGT-3′ and 5′-TACCATTCACGTCTGCTCCA-3′ for Jmjd2a; 5′-CGGGGCTTTTCACACAGTAC-3′ and 5′-GTACAGGGAGCCACTGATGT-3′ for Jmjd2b; 5′-GTCCCCTAAATCCCAGCTGT-3′ and 5′-TAGCACTGTCTTG GCTTCCA-3′ for Jmjd2c; 5′-GGTGGTTGAAGAGCTGGGTA-3′ and 5′-TCACTT CCCTGGATGCATCA-3′ for Setdb1; and 5′-CAACCTTGATGAGCGACTAC-3′ and 5′-CCATTCGGGTACTCTCCA-3′ for SUV39H1; 5′-AAGAGGTGGTGA A GCAGGCATC-3′ and 5′-CGAAGGTGGAAGA GTGGGAGTTG-3′ for Gapdh; and 5′-GCTGCCATCTGTTTTACGG-3′ and 5′-TGACTGGTGCCTGATGAA CT-3′ for L32.

## Chromatin immunoprecipitation (ChIP)-qPCR analyses.

ChIP-qPCR assays were carried out as previously described[38,63] using a ChIP-IT Express Kit (102026; Active Motif). A total of 40–50 mg of hippocampal tissues obtained from 5 mice for each group were minced with a razor blade, and incubated in 7 ml of 1% for-maldehyde (F1635; Sigma-Aldrich) in 1X PBS for 8 min to cross-link proteins and DNA in tissue sample. The cross-linking reaction was stopped by 5 ml of 0.125 M glysine for 5 min, and the tissues were homogenized using a Dounce homogenizer (357542; Wheaton, Millville, NJ, USA). The homogenate was centrifuged at 1250 $g$ for 5 min, and the pellet was resuspended in 500 μl of lysis buffer containing a proteinase cocktail (0.5×, final) and phenylmethanesulfonyl fluoride (PMSF) (0.5 mM, final), and then incubated on ice for 30 min. The lysates then were homo-genized using a Dounce homogenizer and washed twice with 1X PBS. The homogenates were centrifuged at 1250 $g$ and 4 °C for 10 min, and the pellet con-taining fixed chromatin was resuspended in 350 μl of shearing buffer containing proteinase inhibitors and PMSF.

The fixed chromatin was sheared into 200–800 bp fragments by sonication on ice using an Epishear probe sonicator (Active Motif) with 20-s pulses at 50-s intervals at 35% power, with the process repeated 20 times. The sheared chromatin samples were centrifuged at 22,000 $g$ for 10 min, and the supernatant was saved for immunoprecipitation.

To prepare sheared chromatin samples from HT22 cells, cells grown to 70–80% confluence in a 100-mm dish were treated for 24 h with corticosterone or $H_2O_2$ in DMEM containing 1% FBS. After washing with 1X PBS, cells were treated with 1% formaldehyde in 1X PBS for 10 min to cross-link proteins and DNA. The cross-linking reaction was stopped with glycine, as above. Cells were washed with 1X PBS and collected with 500 μl of lysis buffer containing a proteinase cocktail (final 0.5×) and PMSF (final 0.5 mM). The lysates were homogenized using a Dounce homogenizer and centrifuged at 1250 $g$ and 4 °C for 10 min. The pellet containing fixed chromatins was resuspended in 350 μl of shearing buffer, and the fixed chromatins were sheared to 200–800 bp fragments by sonication, as above. After centrifugation, the supernatant containing the sheared chromatin was saved.

The DNA in the sheared chromatin samples was quantified using NanoDrop (Thermo Fisher Scientific). Successful chromatin shearing was confirmed by agarose gel electrophoresis.

For immunoprecipitation reactions, 10 μg of each sheared chromatin was mixed with 2 μg of primary antibody, 20 μl of Protein G magnetic beads, 10 μl of 10X ChIP buffer 1 from the ChIP Assay Kit (Active Motif), and 1 μl of proteinase inhibitor cocktail in a volume of 100 μl, and incubated at 4 °C overnight.

Antibodies used included SUV39H1 (05–615; RRID:AB_2196724, Millipore), acH3K9 (ab10812; RRID:AB_297491, Abcam), dimeH3K9 (ab1220; RRID: AB_449854, Abcam), and trimeH3K9 (ab8898; RRID:AB_306848, Abcam).

The immunoprecipitation products were washed with ChIP buffer I, and then twice with ChIP buffer II on a magnetic rack (Active Motif). After the final wash, the beads were resuspended in 50 μl of elution buffer and incubated at room temperature for 15 min with intermittent agitation. Then, 50 μl of reverse cross-linking buffer was added to each sample. The supernatant containing eluted chromatins was carefully taken on a magnetic rack and incubated at 95 °C for 15 min. The input DNA (10 μl) from the saved sheared chromatin was also subjected to reverse cross-linking after mixing with 88 μl of ChIP buffer II and 2 μl of 5 M NaCl and incubated at 95 °C for 15 min. The reacted mixtures were added to 1 μg of proteinase K and incubated at 37 °C for 1 h, followed by adding 2 μl of proteinase K stop solution. The resulting reaction was regarded as immunoprecipitated DNA and used for real-time PCR.

The real-time PCR reaction was performed with 4 μl of the immunoprecipitated DNA, 10 μl of 2X iQ™ SYBR Green Supermix (Bio-Rad Laboratories), and 1 μl each of 5 pmol/μl forward and reverse primers in a volume of 20 μl using the CFX 96 Real-Time PCR System Detector (Bio-Rad Laboratories).

One-tenth of the input DNA was used for quantitative PCR to control the relative amounts of DNA fragments in immunoprecipitation and normalization for quantification.

Real-time PCR was carried out using the following primers: 5′- GAACTGTAG AGAGGCCCCAG-3′ and 5′-AGTGGTTTCTGCTGCTTGTG-3′ for p47phox P1, 5′-AACACCAGAAGAGGGCATCA-3′ and 5′- GGGGCCTCTCTACAGTTC TG-3′ for p47phox P2, 5′-GAGGATTAATTGGCAGGAATGC-3′ and 5′-GCA GCATTCCTCCAGTGTTT-3′ for gp91phox P1, 5′-CACTTCATCCAGGGGC

AGAG-3′ and 5′-AGGTCAAGGTCTGGCTTAGC-3′ for gp91phox P2, 5′- CGT GTCACAGAGCCTCCTAT-3′ and 5′- TAGACAGGAACAGGCTGAGC-3′ for p67phox P1, 5′-GTTGGCAGACACATGCAGAA-3′ and 5′- ATAGGGAGGCT CTGTGACACG-3′ for p67phox P2.

## Stereotaxic injections of siRNA.

Stereotaxic injection of siRNA was carried out as previously described[38]. Mice were anesthetized by intraperitoneal injection of a mixture (3.5:1) of ketamine hydrochloride (50 mg/ml) and xylazine hydrochloride (23.3 mg/ml) at a dose of 2.5 μl/g body weight. siRNA-control (SN-1012), siRNA-Ppp2ca (#1411897, NM_019411.4), and siRNA-SUV39H1 (#1433203, NM_011514.1) were purchased from Bionner Co. (Daejeon, Korea). One volume of each siRNA (50 ng/μl) was mixed with 2.5 volumes of Neurofect transfection reagent (T800075; Genlantis, San Diego, CA, USA) and 0.5 volume of 50% sucrose. The siRNA mix was incubated for 20 min prior to stereotaxic injection. A volume of 1.8 μl of the siRNA mix (7.5 ng/μl) was injected into each CA3 region (stereo-taxic coordinate: AP, −1.9; ML, ± 2.1; DV,−2.1 mm) at a speed of 0.2 μl/min using a stereotaxic injection system (Vernier Stereotaxic Instrument, Leica Biosystems, Wetzlar, Germany) and a Hamilton syringe with a 30 G needle. After injection, mice were kept on a warm pad until they were awakened, and housed afterward in home cages. Behavioral tests were performed between 48 h and 72 h after siRNA injection. The subjects were sacrificed for tissue preparation at the indicated time point.

## Behavioral analyses.

Behavioral tests were carried out as previously described[38,62]. All tests were performed during the light cycle (9 a.m. to 4 p.m.). For acclimation, mice were brought to the behavior testing room 30 min prior to the start of each behavioral test. In the testing room, the background sound was masked with 65 dB of white noise generated by a white noise generator (HDT Korea, Seoul, Korea). The testing room was lit with indirect illumination by 20 lx for the U-shaped two-choice field and 250 lx for the tail suspension test (TST) and forced swimming test. All behavioral tests were recorded with a computerized video tracking system (SMART, PanLab, Spain) or a webcam recording system (HD Webcam C210, Logitech, USA).

Behavioral tests among groups and within a group were conducted in a randomized fashion and/or in an alternative manner with respect to a test order and position in the U-shaped two-choice field (e.g., left vs. right). Immobility time in the TST and FST for different groups was recorded in a blind manner. Behavioral tests were performed in the order of the sociability test, TST and FST on three days.

## Sociability test.

The sociability test using a U-shaped two-choice field was per-formed as previously described[62]. The U-shaped two-choice field was a modified open field (45 cm × 45 cm) partially divided into two-spaces with a partitioning wall (45 cm wide × 40 cm height) to the central point, so that the open field space appeared as a U-shaped two-choice field, each side containing a closed square and an open square[62]. A circular grid cage (12 cm diameter × 33 cm height) was placed at each side of the closed squares. A grid cage carrying a social target was placed at a closed square, while an empty cage was placed at the other closed square. The side containing a social target was defined as the target field, and the other side con-taining an empty cage was called as the non-target field.

For acclimation, subject mice were individually introduced to the middle of the U-shaped field and allowed to freely explore for 5 min. After acclimation, mice were returned to their home cage for 1 min, while a social target was loaded in a grid cage at a side. Subject mice were then exposed to the U-shaped field containing a social target on one side and an empty cage on the other and allowed to freely explore for 10 min. Normal B6 mice were used as a social target, and each target mouse was served as a social target 3 or 4 times. All animals, including subjects and social targets, were purchased at the same time. Upon arrival, they were grouped and housed in pairs in standard plastic home cages until use. Social interaction time and distance in the U-shaped two-choice field were recorded using a computerized video tracking system (SMART, PanLab, Spain).

Sociability index was calculated as [the percent time spent in the target zone minus the percent time in the non-target zone] divided by [the percent time spent in the target zone plus the percent time in the non-target zone], as previously described[62].

## Tail suspension test.

The TST was carried out as previously described[38,62]. Mice were individually suspended by fixing their tails to the ceiling of a white wooden box (30 × 45 × 50 cm) using adhesive tape and holding them 50 cm above the surface of a table. The cumulative immobility time of each mouse was counted during the first 6 min period. The TST was recorded using a webcam recording system (HD Webcam C210, Logitech, USA).

## Forced swimming test.

The forced swimming test was carried out as previously described[38,62]. Mice were placed individually in a Plexiglas cylinder (15 cm in diameter × 27 cm in height) filled with water at 24 °C at a depth of 14 cm. Subject mice were individually put in a Plexiglas cylinder for 6 min and their escape-related mobility behavior was measured during the last 5 min. Immobility time was

defined as summed time displaying floating behavior with all limbs motionless. The FST was recorded using a webcam recording system.

**K-Means clustering**. SPSS Statistics 25 software (IBM SPSS Statistics, NY, USA) was used in k-Means cluster analyses. K-means clustering is an unsupervised machine-learning algorithm that classifies individuals into similar groups. Depending on the $k$ values, all data points are assigned to the nearest cluster.

**Statistical analyses**. GraphPad PRISM 6.0 software (GraphPad Software. Inc., CA, USA) was used for statistical analyses. Two-sample comparisons were carried out using the Student's $t$ test, while multiple comparisons were made using one-way analysis of variance (ANOVA) followed by a Newman–Keuls post hoc test or two-way ANOVA followed by a Bonferroni post hoc test. The details of the statistical analysis were provided in Supplementary Table 1. All data are presented as mean ± standard error of the mean (SEM), and statistical significance was accepted at the 5% level.

**Reporting summary**. Further information on research design is available in the Nature Research Reporting Summary linked to this article.

## Data availability
The raw data of Western blots images are provided as Supplementary Data 1; the source data for the graphs in the main figures and supplementary figures are included in Supplementary Data 2; the details of the statistical analysis of all figures are provided in Supplementary Data 3. All other relevant data of the current study will be available from the corresponding author on reasonable request.

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

## Acknowledgements

This research was supported by a grant (2018R1A2B2001535) from the Ministry of Science, ICT and Future Planning, Republic of Korea.

## Author contributions

J.E.L. and P.L.H. developed the concept and initial experiments, and designed the detailed experiments; J.E.L., H.J.K., J.C., and J.S.S. carried out experiments; J.E.L. and P.L.H. analyzed and interpreted collected data, and wrote the manuscript.

## Competing interests

The authors declare no competing interests.
