## [Peer review file · Communications Biology]

Reviewers' comments:

Reviewer #1 (Remarks to the Author):

This study greatly expands on the proposition that psychological stress produces depressive-like behaviors via a process of stress-induced elevated glucocorticoids that epigenetically promote oxidative stress in hippocampal neurons, which disrupts neuronal activity in this limbic structure. Further, the process of aging amplifies the pathway, making stress more consequential in older animals. The data supply evidence for a string of events whereby GCs elevate Ppp2A, which downregulates p-AMPK and SUV39H1 activities, which elevates NADPH subunits p47phox and p91phox, which elevates ROS activity. Data are collected from in vivo experiments in WT and p47phox knockout mice and from HT22 cell lines. All of the evidence, which includes multiple pharmacological, biochemical, and histological assays plus three behavioral tasks, support the findings. Causality is demonstrated by several methods, including siRNA knockdown, AMPK (AICAR, Compound C, and RA) manipulation, and corticosterone manipulation (Cort or RU486). The weakest part of the study is the inability to definitively localize the pathway to neurons in field CA3 of the hippocampus. Most of the aforementioned tools have body-wide effects, and specificity is implied by data from the cell lines, but that is not completely satisfactory. The other positive assignment is by the effective application of siRNA to the CA3 region bilaterally, but that is only done twice (Figs 4 and 6) with only a few outcomes measures taken.

Points:

1. How does p-AMPK negatively regulate p47phox, as indicated by the negative correlation shown in Fig. 3? This relationship should be explained more clearly. The citations about regulation are from studies in neutrophils and endothelia. What is known about neurons?
2. The story does not well take into account the off-target actions of the reagents. The P47phox knockout is body-wide with many peripheral consequences. RA does not penetrate from the periphery into the brain DOI <https://doi.org/10.1038/s41598-019-45168-1>, so its action to counter depressive effects in vivo is unexplained. Focus on the hippocampus, more precisely the CA3 region, is questionable because that region's involvement in the behaviors measured is not that crucial. Also, ROS levels are elevated in all cells, not just neurons, and in all areas, not just CA3 or the hippocampus. The DHE results are misleading because the incident wavelength was too high to ensure specificity of the fluorescence outcome measure. Microscopy of marker co-registration does not show solid evidence for that, it is more like a few cells showing co-registration.
3. It is not clear from the Methods how the behavioral tests were conducted. Presumably each animal underwent one test per day for three successive days, but that is not stated, nor is the order of the tests stated. A concern about the test selection is that TST and FST are typically not done in the same animal because they are both stressful procedures, and the effects of one test can negate the ability to get reliable data on the second test (e.g., see "Chapter 27, Animal Models of mood disorders," by Sillaber Holsboer and Wotjak in Charney and Nestler (eds), "Neurobiology of Mental Illness"). Furthermore, the two tests measure the same kind of behavior, so it is not considered appropriate to do both tests.
4. The finding that old mice have higher basal Cort levels than young mice is surprising but not discussed, and it should be. Other researchers do not find this in aging (e.g., <https://doi.org/10.3389/fnagi.2014.00013>, <https://www.ncbi.nlm.nih.gov/pubmed/20416975>, DOI: 10.1055/s-2007-972302, <https://doi.org/10.1371/journal.pone.0136112>)
5. Line 155 on p. 7: dihydroergotamine should be dihydroethidium

Reviewer #2 (Remarks to the Author):

In this work the authors study the underlying mechanisms of the changes in differential sensitivity to stress observed in aging brain. The authors observe an increase in oxidative stress in the hippocampus of aged mice, which they link to an increase in the levels of the enzyme NADPH oxidase and an increase in depression behavior in these animals. Specifically, they authors identify p47phox, one of the subunits of the enzyme as an important regulatory factor in this process. The authors link the control of these factors during aging, together with behavioral effects, with AMPK

signaling and identify the phosphatase PPP2a and the histone methyltransferase Suv39h1 as downstream effectors of this signaling on the expression of NADPH oxidase subunits. This is a very relevant subject and should be of interest to a wide-range of researchers from different fields. The molecular basis of brain declining function during aging are not well understood. In this sense, the work represents a novel and relevant step forward by linking the previously reported increase of oxidative stress in hippocampus regions of aging brains with an increase in stress-induced depression. The authors have done an impressive amount of work and in general their claims are well demonstrated. However, there are some issues, scientific and technical that should be addressed:

1) The main issue is regarding the epigenetic mechanism proposed to explain the control of NADPH:

a) Considering the amount of epigenetic enzymes that have been linked to AMPK signaling, like for instance polycomb H3K27me3 enzyme Ezh2 or the NAD⁺-dependent deacetylases SIRT1 and SIRT7, why do the authors chose these factors including Suv39h1? Is there any specific reason for that? Is H3K27me3 also involved?

b) Is HDAC2 also present in p47phox and gp91phox? Has any relevant effect there?

c) In figure 5-6 (for instance figure 6e-n) authors test the expression of these NADPH oxidase genes under modulation of these factors but never test the actual protein levels. A western-blot should be included to validate their model.

d) The authors correlate the expression of these genes with H3K9me3 but do not demonstrate whether is important in this control. Would expression or inhibition of H3K9me3 demethylases tested have any significant effect on the expression of these genes?

2) In fig 3c, what are the total levels of AMPK? Are they altered? The authors should include a loading control to assess these changes.

3) Figure 3d is a concern. The authors state "Immunofluorescence staining revealed that p-AMPK and p47phox were co-expressed at the single-cell level in pyramidal neurons of the hippocampus, in which expression levels of p47phox were negatively correlated with those of p-AMPK (Fig. 3d,e)." However, in addition to the specific regions pointed to by the authors, this antagonism is not clear in the vast majority of the cells shown there. Unless the authors show more convincing data, this is an overstatement.

4) In the same line, I am not sure I understand what represents the experiment in figure 3e or 5n. Are these quantifications of the IF signals? Or expression measurements? I have not been able to find any clear description about these experiments in the figure legends or the main text.

5) The effect of GC on p-AMPK/AMPK is very small and it's difficult to believe that is biologically relevant, particularly when compared with the differences observed in aging (Figure 3c). This is somehow contradictory with the clear effect on p47phox expression and the compensation observed with the AMPK activator AICAR (figure 3g). How do the authors explain these divergences? Maybe a big part of the CG effect on p47phox is not really dependent on AMPK, although it may be compensated by its activation?

MINOR issues:

1) Fig 4c. The white point in the legend does not correspond to the color of the points in the graph.

2) In the text DHE is presented as dihydroergotamine and dihydroethidium

We have followed the editor's suggestion and now provide a substantial amount of new data to address the referees' very constructive suggestions. Indeed, we believe the paper has been greatly strengthened by this revision. A detailed summary of the revisions is given below.

Reviewers' comments:

Reviewer #1 (Remarks to the Author):

This study greatly expands on the proposition that psychological stress produces depressive-like behaviors via a process of stress-induced elevated glucocorticoids that epigenetically promote oxidative stress in hippocampal neurons, which disrupts neuronal activity in this limbic structure. Further, the process of aging amplifies the pathway, making stress more consequential in older animals. The data supply evidence for a string of events whereby GCs elevate Ppp2A, which downregulates p-AMPK and SUV39H1 activities, which elevates NADPH subunits p47phox and p91phox, which elevates ROS activity. Data are collected from in vivo experiments in WT and p47phox knockout mice and from HT22 cell lines. All of the evidence, which includes multiple pharmacological, biochemical, and histological assays plus three behavioral tasks, support the findings. Causality is demonstrated by several methods, including siRNA knockdown, AMPK (AICAR, Compound C, and RA) manipulation, and corticosterone manipulation (Cort or RU486). The weakest part of the study is the inability to definitively localize the pathway to neurons in field CA3 of the hippocampus. Most of the aforementioned tools have body-wide effects, and specificity is implied by data from the cell lines, but that is not completely satisfactory. The other positive assignment is by the effective application of siRNA to the CA3 region bilaterally, but that is only done twice (Figs 4 and 6) with only a few outcomes measures taken.

1. How does p-AMPK negatively regulate p47phox, as indicated by the negative correlation shown in Fig. 3? This relationship should be explained more clearly. The citations about regulation are from studies in neutrophils and endothelia. What is known about neurons?

→ The reviewer requested to explain the mechanism by which p-AMPK regulates p47phox. To address reviewer's comment, we carried out a new experiment to investigate the relationship between p-AMPK and p47phox. We now show that AMPK activation with AICAR in HT22 cells increased p-CREB level, whereas its inhibition with compound C (CC) decreased p-CREB (Fig. 5l,m). Furthermore, siRNA-mediated inhibition of CREB decreased SUV39H1 expression while increasing p47phox expression (Fig. 5n). These results are consistent with the finding that siRNA-mediated inhibition of SUV39H1 upregulated the expression of p47phox and gp91phox (Fig. 6a,b). Together, these results indicate that p-AMPK regulates p47phox expression via p-CREB and SUV39H1. We include these new data in the revised manuscript.

Fig. 5l-n

2. The story does not well take into account the off-target actions of the reagents. The P47phox knockout is body-wide with many peripheral consequences. RA does not penetrate from the periphery into the brain DOI <https://doi.org/10.1038/s41598-019-45168-1>, so its action to counter depressive effects in vivo is unexplained. Focus on the hippocampus, more precisely the CA3 region, is questionable because that region's involvement in the behaviors measured is not that crucial. Also, ROS levels are elevated in all cells, not just neurons, and in all areas, not just CA3 or the hippocampus. The DHE results are misleading because the incident wavelength was too high to ensure specificity of the fluorescence outcome measure. Microscopy of marker co-registration does not show solid evidence for that, it is more like a few cells showing co-registration.

→ The reviewer raised a legitimate concern on interrelated issues regarding off-target actions of the reagents:

(i) Concerning p47phox KO mice and the role of the CA3 hippocampus in regulation of depressive behaviors;

→ p47phox KO mice have a whole body knockout of p47phox, and therefore, as reviewer pointed, p47phox KO mice do not specify the role of p47phox in the hippocampus. However, viral vector-assisted shRNA-mediated knockdown of p47phox within the hippocampus produced anti-depressive behaviors (ref. S1). Furthermore, siRNA-mediated knockdown of SUV39H1 in a CA3 subregion of the hippocampus produced anti-depressive effects (ref. S2). It is in line with those results that siRNA-mediated inhibition of Ppp2ca in the CA3 subregion produced anti-depressive effects (Fig. 4g-l). Therefore, these results suggest that the CA3 subregion or the hippocampus function as a critical neuronal node in the neural networks regulating depressive behaviors (ref. S3). We speculate that the intervention of the CA3 subregion or the hippocampus can sensitively modify the activity of the neural networks regulating depressive behaviors. We believe that this interpretation does not necessarily underscore the importance of other brain regions in the neural networks that regulate depressive behaviors.

(ii) Concerning RA availability in the brain;

→ As the reviewer concerned, the bioavailability of orally fed RA is low. Only 0.9 - 1% of RA given orally was absorbed into the blood (ref. S4, ref. S5), whereas the orally fed RA (150 μmol of RA/kg as a component of 600 mg of herbal tea/kg administered) was not detected in the brain (ref. S5). When RA was intraperitoneally administered in rats, 0.4% of the dose administered was detected in the brain 30 min after injection (ref. S5). Although if this brain delivery rate can be applicable to mice, the amount of RA that can reach the brain in mice with 25 g of body weight, after injected with 30 mg/kg of RA, will be 3 μg , which is equivalent to 238 μM when we assumed 3 μg is dissolved in 35 μL of CSF (ref. S6). To overcome the complexity of biological responses in different animal species and the possible RA metabolism that continuously occurred peripherally and centrally, we carried out a new experiment to directly infuse RA into the hippocampus. RA (200 μM , 1.8 μL) was infused into the CA3 subregion of the hippocampus via a preimplanted cannula, five times in every 12 h in 2.5 days. The infusion into CA3 region suppressed Ppp2ca expression, increased SUV39H1 and p-AMPK levels, and decreased p47phox and gp91phox levels (Supplemental Fig. 8). These results are in line with the signaling responses in the hippocampus of mice injected intraperitoneally with RA (Fig. 7g-k; Supplemental Fig. 9a-h) and in HT22 cells treated with RA (Fig. 7a-f). We include these new data in Supplemental Fig. 8 in the revised manuscript.

Supplemental Fig. 8

(iii) Concerning that ROS levels are elevated in all cells, not just neurons, and in all areas, not just CA3 or the hippocampus.

→ We carried out new experiment to examine where ROS was accumulated in the hippocampus. Immunofluorescence staining followed by DHE reaction indicated that DHE-reactive ROS was accumulated in NeuN-positive cells, and in some non-NeuN-positive cells in the CA1 and CA3 regions of the hippocampus (Supplemental Fig. 1). We include these new data in the revised manuscript.

Supplemental Fig. 1

3. It is not clear from the Methods how the behavioral tests were conducted. Presumably each animal underwent one test per day for three successive days, but that is not stated, nor is the order of the tests stated. A concern about the test selection is that TST and FST are typically not done in the same animal because they are both stressful procedures, and the effects of one test can negate the ability to get reliable data on the second test (e.g., see “Chapter 27, Animal Models of mood disorders,” by Sillaber Holsboer and Wotjak in Charney and Nestler (eds), “Neurobiology of Mental Illness”). Furthermore, the two tests measure the same kind of behavior, so it is not considered appropriate to do both tests.

→ We now include a more detailed procedure of the behavioral tests in the Methods in the revised manuscript. Each animal underwent one test per day for three successive days in the order of the sociability test (U-box), TST, and FST. Aged mice (18 M) have ~35 g of a body weight, which makes them hard to react in a suspension condition by the tail, so that aged animals underwent the sociability test and FST, but TST.

→ In the experiments involving stereotaxic siRNA injection (Fig. 4g), siRNA-mediated knockdown of target genes significantly decayed after ~3 days of injection (ref. S7, ref. S8). Therefore, for those mouse groups, the behavioral tests were performed on the second day after siRNA injection (Fig. 4g). Mice injected with siRNA-Ppp2ca showed a recovery of behavioral deficits not only in the first test (TST), but also in the following

test (FST). Furthermore, behavioral performance of siRNA-injected animals in Fig. 4g was comparable to that of uninjected animals that were tested for two or three successive days (eg., Supplemental Fig. 5d-f, j-l; Supplemental Fig. 9i,k,l). These results suggest that the behavioral test paradigm used for siRNA-injected animals was not affected by cumulative behavioral test effects.

→ TST and FST are known to measure hopelessness and helplessness behaviors. However, several lines of evidence suggest that TST and FST measure somewhat different aspects of hopelessness and helplessness behaviors. When mice are placed in the TST and FST consecutively, some individuals exhibit increased immobility in the first test, while displaying control-like immobility in the second test, suggesting that a dissociation of behavioral responses can be found in individual animals sequentially placed in the FST and FST, as indicated in Figure S1, which is a redrawing of individual animals of Supplemental Fig. 9k and Fig. 9l into the TST and FST matrix using the values of z-score.

Figure S1

→ We agree that the precedent behavioral test can affect the following behavioral tests given in the same animals. Particularly, if the same test (eg., FST) is repeatedly given in the same animals, the precedent behavioral test affects the following behavioral tests over time. On the contrary, if different behavior tests (eg., sociability test, TST and FST) are given in the same animals after a certain time interval, following behavioral tests are practically unaffected by the precedent behavioral test.

4. The finding that old mice have higher basal Cort levels than young mice is surprising but not discussed, and it should be. Other researchers to not find this in aging (e.g., <https://doi.org/10.3389/fnagi.2014.00013>, <https://www.ncbi.nlm.nih.gov/pubmed/20416975>, DOI: 10.1055/s-2007-972302, <https://doi.org/10.1371/journal.pone.0136112>)

→ Thanks. Aged mice had increased basal but not stress-induced levels of corticosterone relative to those in young mice (Fig. 1a). The result of the increased basal corticosterone levels in aged mice is consistent with the previous reports (ref. S9, ref. S10, ref. S11, ref. S12). These results suggest that the stress system in aged mice is not tightly controlled under normal physiological conditions as much as young mice and also do not effectively respond to incoming stress compared with young mice. We include this discussion in the Discussion section in the revise manuscript.

→ As pointed by the reviewer, there are studies reporting that Cort levels were not increased in aged rats (e.g, ref. S13). Regarding that corticosterone levels are sensitively affected by various experimental factors, including handling stress, blood collection time in a day, mouse/rat strains/species, and animal room condition, and careful and systematic analyses are required to elaborate what makes such a discrepancy.

5. Line 155 on p. 7: dihydroergotamine should be dihydroethidium

→ Thanks. We corrected the mistake in the revised manuscript.

Reviewer #2 (Remarks to the Author):

In this work the authors study the underlying mechanisms of the changes in differential sensitivity to stress observed in aging brain. The authors observe an increase in oxidative stress in the hippocampus of aged mice, which they link to an increase in the levels of the enzyme NADPH oxidase and an increase in depression behavior in these animals. Specifically, they authors identify p47phox, one of the subunits of the enzyme as an important regulatory factor in this process. The authors link the control of these factors during aging, together with behavioral effects, with AMPK signaling and identify the phosphatase PPP2a and the histone methyltransferase Suv39h1 as downstream effectors of this signaling on the expression of NADPH oxidase subunits.

This is a very relevant subject and should be of interest to a wide-range of researchers from different fields. The molecular basis of brain declining function during aging are not well understood. In this sense, the work represents a novel and relevant step forward by linking the previously reported increase of oxidative stress in hippocampus regions of aging brains with an increase in stress-induced depression. The authors have done an impressive amount of work and in general their claims are well demonstrated. However, there are some issues, scientific and technical that should be addressed:

1. The main issue is regarding the epigenetic mechanism proposed to explain the control of NADPH:

a) Considering the amount of epigenetic enzymes that have been linked to AMPK signaling, like for instance polycomb H3K27me3 enzyme Ezh2 or the NAD⁺-dependent deacetylases SIRT1 and SIRT7, why do the authors chose these factors including Suv39h1? Is there any specific reason for that? Is H3K27me3 also involved? → As the reviewer pointed, Ezh2, SIRT1 and SIRT7 have a role in epigenetic regulation of AMPK in various types of cells. Our new experiment indicate that the expression of Ezh1, but not Ezh2, increased in the hippocampus of aged mice, whereas the expression of Sirt1, but not Sirt7, decreased (Figure S2a,b). Ezh1 normally reduces the expression of target genes, and therefore increased expression of Ezh1 in aged mice does not likely play a role in the upregulation of the p47phox and gp91phox.

The expression of the histones deacetylase Sirt1 decreased in the hippocampus of aged mice (Figure S2a,c). Sirt1 normally makes the promoter regions less accessible to transcription factors. Therefore, it is possible decreased expression of Sirt1 plays a role to increase p47phox and gp91phox expression in aged mice, although we did not go on studying Sirt1 effects in the present study. We included Sirt1 and Sirt7 data in Fig. 5a and d in the revised manuscript.

Figure S2

b) Is HDAC2 also present in p47phox and gp91phox? Has any relevant effect there?
 → HDAC2 facilitates to form a transcription repressor complex, and therefore, reduced expression of HDAC2 can increase the expression of target genes. Indeed, our new experiment showed that siRNA-Hdac2 injection into CA3 subregion produced an increase in the expression of the p47phox and gp91phox (Figure S3). However, because HDAC2 expression increased in the hippocampus of aged mice (Fig. 5a,c), we did not count on HDAC2 to regulate brain aging in the present study.

Figure S3

c) In figure 5-6 (for instance figure 6e-n) authors test the expression of these NADPH oxidase genes under modulation of these factors but never test the actual protein levels. A western-blot should be included to validate their model.

→ Our new immunohistochemical analysis indicates that SUV39H1 protein expression was downregulated in the hippocampus of aged mice compared with young mice (Fig. 5f,g). A further quantitative analysis indicates that the expression levels of SUV39H1 were negatively correlated with the expression of p47phox protein at a single cell level in the CA1 CA3 and DG regions of the hippocampus (Supplemental Fig. 6f,g). We include these new data in Fig. 5f,g and Supplemental Fig. 6f,g in the revised manuscript.

Fig. 5f,g

Supplemental Fig. 6f,g

d) The authors correlate the expression of these genes with H3K9me3 but do not demonstrate whether is important in this control. Would expression or inhibition of H3K9me3 demethylases tested have any significant effect on the expression of these genes?

→ SUV39H1 and Setdb1 are H3K9 methyltransferases, and Jmjd2a, Jmjd2b, Jmjd2c are H3K9 demethylases. Among these factors, SUV39H1 was significantly decreased in the hippocampus of aged mice, whereas others are not significantly (Fig. 5a,b). siRNA-mediated inhibition of SUV39H1 in the CA3 subregion increased the expression of the p47phox and gp91phox (Fig. 6a,b). Furthermore, the levels of SUV39H1, trimeH3K9, and dimeH3K9 bound to the promoter region of the p47phox and gp91phox were downregulated, supporting the role of H3K9me3 in the upregulation of the p47phox and gp91phox (Fig. 6e-n).

2. In fig 3c, what are the total levels of AMPK? Are they altered? The authors should include a loading control to assess these changes.

→ Thanks. We repeated Western blotting analysis of AMPK and p-AMPK, and now present improved data with b-actin control in Fig. 3c in the revised manuscript.

Fig. 3c

3. Figure 3d is a concern. The authors state "Immunofluorescence staining revealed that p-AMPK and p47phox were co-expressed at the single-cell level in pyramidal neurons of the hippocampus, in which expression levels of p47phox were negatively correlated with those of p-AMPK (Fig. 3d,e)." However, in addition to the specific regions pointed to by the authors, this antagonism is not clear in the vast majority of

the cells shown there. Unless the authors show more convincing data, this is an overstatement.

→ The reviewer raised a question about the antagonistic relationship between p-AMPK and p47phox is not clear. We revised the photomicrographs with a higher magnification and adding selected labels of the neurons expressing p-AMPK and p47phox, each with a contrasting level. For quantification, we randomly selected neurons in CA3 subregion and read signal intensity levels of p-AMPK and p47phox at individual cells using a MetaMorph image analysis program (Figure S4). The intensity values of p-AMPK and p47phox expression levels at individual cells were plotted. The combined resulting data indicate that the expression levels of p-AMPK and those of p47phox were, at the single-cell level, negatively correlated with $r^2 = 0.218$ and $p = 0.0007$, as summarize in Figure 3e. Without quantification and plotting the correlation, it is hard to read whether there is a negative correlation between p-AMPK and p47phox. We see that there are many cells do not show such antagonistic expression. However, if there is no correlation, this quantification data should give much lower r^2 value and higher p value. The detailed quantification procedure was included in the Methods in the revised manuscript.

Figure S4

4. In the same line, I am not sure I understand what represents the experiment in figure 3e or 5n. Are these quantifications of the IF signals? Or expression measurements? I have not been able to find any clear description about these experiments in the figure legends or the main text.

→ The reviewer raised the same question about the antagonistic relationship between p-AMPK and p47phox (Fig 3e), and between SUV39H1 and p47phox (Supplemental Fig. 6g). As stated above, to quantify expression relationship between p-AMPK and p47phox and between SUV39H1 and p47phox, we randomly selected neurons in CA3 subregion and read signal intensity levels of p-AMPK and p47phox, and of SUV39H1

and p47phox, at individual cells using a MetaMorph image analysis program. The intensity values of p-AMPK and p47phox, and of SUV39H1 and p47phox expression levels at individual cells were plotted. The resulting data indicate that the expression levels of p-AMPK and those of p47phox and of SUV39H1 and p47phox were, at the single-cell level, negatively correlated with $r^2=0.218$ and $p=0.0007$ in Fig. 3e and $r^2=0.1343$ and $p=0.0464$ in Supplemental Fig. 6g, respectively. We see that there are many cells do not show such antagonistic expression, suggesting that there might be other factors that regulate p47phox expression. However, if there is no correlation, this quantification data should give much lower r^2 value and higher p value. The detailed quantification procedure was included in the Methods in the revised manuscript.

5. The effect of GC on p-AMPK/AMPK is very small and it's difficult to believe that is biologically relevant, particularly when compared with the differences observed in aging (Figure 3c). This is somehow contradictory with the clear effect on p47phox expression and the compensation observed with the AMPK activator AICAR (figure 3g). How do the authors explain these divergences? Maybe a big part of the CG effect on p47phox is not really dependent on AMPK, although it may be compensated by its activation?

→ As pointed by the reviewer, the GC-induced change in p-AMPK/AMPK ratio examined in HT22 cells was significant, but its change was relatively small (Fig. 3f), compared to the big shift in the brain of aged mice. The GC-induced increase in the expression of p47phox became greater when GC was treated at higher concentrations and for longer durations (Supplemental Fig. 2c,d). Furthermore, GC-induced increase in the expression of p47phox was markedly suppressed by increasing AMPK activity with AICAR (Fig. 3g). These results are consistent with the notion that GC decreases p-AMPK/AMPK level, which in turn downregulates p47phox expression. Regarding these results, it is possible that the small change in the GC-induced p-AMPK/AMPK ratio in HT22 cells was partly due to the concentration and duration of GC used in culture. However, we do not exclude the possibility that there is an unidentified parallel pathway mediating GC effects on p47phox.

Fig. 3f

Fig. 3g

MINOR issues:

1) Fig 4c. The white point in the legend does not correspond to the color of the points in the graph.

→ Thanks. We corrected the mistake in Fig. 4c in the revised manuscript.

Fig. 4c

2) In the text DHE is presented as dihydroergotamine and dihydroethidium
 → Thanks. We corrected the mistake in the revised manuscript.

REFERENCE

- S1. QQQ et al., 2012
- S2. QQQ et al., 2019a
- S3. QQQ et al., 2019b
- S4. Konishi, Y. & Kobayashi, S. Transepithelial transport of rosmarinic acid in intestinal Caco-2 cell monolayers. *Biosci Biotechnol Biochem.* **69**, 583-591 (2005).
- S5. Falé, P.L., Madeira, P.J., Florêncio, M.H., Ascensão, L. & Serralheiro, M.L. Function of Plectranthus barbatus herbal tea as neuronal acetylcholinesterase inhibitor. *Food Funct.* **2**, 130-136 (2011).
- S6. Pardridge, W.M. CSF, blood-brain barrier, and brain drug delivery. *Expert Opin Drug Deliv.* **13**, 963-975 (2016).
- S7. QQQ et al., 2014
- S8. QQQ et al., 2015
- S9. Yau, J.L., Olsson, T., Morris, R.G., Meaney, M.J. & Seckl, J.R. Glucocorticoids, hippocampal corticosteroid receptor gene expression and antidepressant treatment: relationship with spatial learning in young and aged rats. *Neuroscience.* **66**, 571-581 (1995).
- S10. Yau, J. L. et al., Chronic treatment with the antidepressant amitriptyline prevents impairments in water maze learning in aging rats. *J Neurosci.* **22**, 1436-1442 (2002).
- S11. Garrido, P., de Blas, M., Del Arco, A., Segovia, G. & Mora, F. Aging increases basal but not stress-induced levels of corticosterone in the brain of the awake rat. *Neurobiol Aging.* **33**, 375-382 (2012).
- S12. Yau, J.L., Noble, J. & Seckl, J.R. 11beta-hydroxysteroid dehydrogenase type 1 deficiency prevents memory deficits with aging by switching from glucocorticoid receptor to mineralocorticoid receptor-mediated cognitive control. *J Neurosci.* **31**, 4188-4193 (2011).
- S13. Buechel, H.M. et al., Aged rats are hypo-responsive to acute restraint: implications for psychosocial stress in aging. *Front Aging Neurosci.* **6**, 13 (2014).

REVIEWERS' COMMENTS:

Reviewer #1 (Remarks to the Author):

The authors have done a lot of work to answer the reviewers' questions, and the manuscript is now much improved.

Many of the references in the ref list are "QQQ et al., 20xx". Many of the missing references are key to the authors' argument, so their omission is disturbing. QQQ also appears throughout the text (methods) to mark vendors, etc.

Do careful reading of new text to ensure that past tenses are correct. New figures and figure legends; make sure what is being shown is named.

Reviewer #2 (Remarks to the Author):

In the new version of the manuscript, the authors have addressed all my concerns. The new data and changes to the manuscript have strengthened the main claims of the work. I only have a couple of minor issues. The first one is that giving the similar behavior of SUV39h1 and SIRT1 in these studies, would be interesting to mention in the manuscript that SIRT1 and SUV39H1 have been reported to interact and cooperate in silencing and heterochromatin formation (Vaquero et al, Nature 2007; Murayama et al., Cell 2008). The second issue is regarding the antagonistic effect observed in immunofluorescence studies between AMPK and p47phox I mentioned in my previous review in points #3 and 4. I agree with the authors that the negative correlation is supported by the new added statistics at single-cell level. However, the fact that this is not observed in all cells, is an interesting observation and should be discussed briefly in the text.

Responses to Reviewers comments

REVIEWERS' COMMENTS:

Reviewer #1 (Remarks to the Author):

The authors have done a lot of work to answer the reviewers' questions, and the manuscript is now much improved. Many of the references in the ref list are "QQQ et al., 20xx". Many of the missing references are key to the authors' argument, so their omission is disturbing. QQQ also appears throughout the text (methods) to mark vendors, etc. Do careful reading of new text to ensure that past tenses are correct. New figures and figure legends; make sure what is being shown is named.

→ Thanks. We now correct and revise all of the pointed parts in the revised manuscript.

Reviewer #2 (Remarks to the Author):

In the new version of the manuscript, the authors have addressed all my concerns. The new data and changes to the manuscript have strengthened the main claims of the work. I only have a couple of minor issues. The first one is that giving the similar behavior of SUV39h1 and SIRT1 in these studies, would be interesting to mention in the manuscript that SIRT1 and SUV39H1 have been reported to interact and cooperate in silencing and heterochromatin formation (Vaquero et al, Nature 2007; Murayama et al., Cell 2008). The second issue is regarding the antagonistic effect observed in immunofluorescence studies between AMPK and p47phox I mentioned in my previous review in points #3 and 4. I agree with the authors that the negative correlation is supported by the new added statistics at single-cell level. However, the fact that this is not observed in all cells, is an interesting observation and should be discussed briefly in the text.

→ Thanks again. We now add a statement on the interaction between SIRT1 and SUV39H1 with suggested references, and also a short discussion about the partial deviation from the negative correlation between AMPK and p47phox I in the Discussion section of the revised manuscript.